# Seipin is required for converting nascent to mature lipid droplets

Huajin Wang[1,2,3], Michel Becuwe[2,3], Benjamin E Housden[4],
Chandramohan Chitraju[2,3], Ashley J Porras[2,3], Morven M Graham[5], Xinran N Liu[5],
Abdou Rachid Thiam[6], David B Savage[7], Anil K Agarwal[8], Abhimanyu Garg[8],
Maria-Jesus Olarte[2,3], Qingqing Lin[2,3], Florian Fröhlich[2,3,9],
Hans Kristian Hannibal-Bach[10], Srigokul Upadhyayula[11,12,13], Norbert Perrimon[4,14],
Tomas Kirchhausen[11,12,13], Christer S Ejsing[10], Tobias C Walther[2,3,14,15]*†,
Robert V Farese Jr[2,3,15]*†

[1]Department of Biological Sciences, Carnegie Mellon University, Pittsburgh, United States; [2]Department of Genetics and Complex Diseases, Harvard T H Chan School of Public Health, Boston, United States; [3]Department of Cell Biology, Harvard Medical School, Boston, United States; [4]Department of Genetics, Harvard Medical School, Boston, United States; [5]Center for Cellular and Molecular Imaging, Department of Cell Biology, Yale School of Medicine, New Haven, United States; [6]Laboratoire de Physique Statistique, École Normale Supérieure, PSL Research University, Université Paris Diderot Sorbonne Paris-Cité, Sorbonne Universités UPMC Univ Paris 06, CNRS UMR 8550, Paris, France; [7]The University of Cambridge Metabolic Research Laboratories, Wellcome Trust-MRC Institute of Metabolic Science, Cambridge, United Kingdom; [8]Division of Nutrition and Metabolic Diseases, Department of Internal Medicine, UT Southwestern Medical Center, Dallas, United States; [9]Molecular Membrane Biology Section, Department of Biology/Chemistry, University of Osnabrück, Osnabrück, Germany; [10]VILLUM Center for Bioanalytical Sciences, Department of Biochemistry and Molecular Biology, University of Southern Denmark, Odense, Denmark; [11]Department of Cell Biology, Harvard Medical School, Boston, United States; [12]Department of Pediatrics, Harvard Medical School, Boston, United States; [13]Program in Cellular and Molecular Medicine, Boston Children's Hospital, Boston, United States; [14]Howard Hughes Medical Institute, Boston, United States; [15]Broad Institute of Harvard and MIT, Cambridge, United States

*For correspondence: twalther@hsph.harvard.edu (TCW); robert@hsph.harvard.edu (RVF)

†These authors contributed equally to this work

**Abstract** How proteins control the biogenesis of cellular lipid droplets (LDs) is poorly understood. Using *Drosophila* and human cells, we show here that seipin, an ER protein implicated in LD biology, mediates a discrete step in LD formation—the conversion of small, nascent LDs to larger, mature LDs. Seipin forms discrete and dynamic foci in the ER that interact with nascent LDs to enable their growth. In the absence of seipin, numerous small, nascent LDs accumulate near the ER and most often fail to grow. Those that do grow prematurely acquire lipid synthesis enzymes and undergo expansion, eventually leading to the giant LDs characteristic of seipin deficiency. Our studies identify a discrete step of LD formation, namely the conversion of nascent LDs to mature LDs, and define a molecular role for seipin in this process, most likely by acting at ER-LD contact sites to enable lipid transfer to nascent LDs.

**eLife digest** Living organisms often store energy in the form of fat molecules called triglycerides. Enzymes in a compartment of the cell called the endoplasmic reticulum catalyze the chemical reactions needed to make these triglycerides. The cell then stores the triglycerides in a different structure called the lipid droplet. Lipid droplets form from the endoplasmic reticulum in an organized manner, but little is known about the cellular machinery that gives rise to lipid droplets.

A protein called seipin is thought to be involved in lipid droplet formation. Seipin resides in the endoplasmic reticulum and a shortage of this protein in cells leads to abnormal lipid droplets – that is, cells often have lots of tiny lipid droplets or a few giant ones. People who lack seipin lose much of their fat tissue and instead store fat in the wrong places, such as the liver.

Now, Wang et al. have studied the seipin protein in insect and human cells grown in the laboratory. The experiments confirmed that cells that lack the seipin protein form lots of tiny dot-like structures containing triglycerides that fail to grow into normal-sized lipid droplets. These lipid droplets have different proteins on their surface, which may impair their ability to store fat. Wang et al. also discovered that in normal cells, the seipin protein is found at distinct spots in the endoplasmic reticulum. This distribution appears to allow seipin to come into contact with the small, newly formed lipid droplets and enable them to grow.

Together these findings suggest that the seipin protein could form part of a molecular machine that allows more triglycerides to be added into newly formed lipid droplets causing the droplets to grow as normal. When seipin is not present the newly formed lipid droplets initially become stuck in a smaller form. As a consequence, a few of these tiny droplets later enter a different cellular pathway of lipid droplet expansion, which turns them into abnormally large lipid droplets.

Future challenges will be to determine precisely how seipin enables newly formed lipid droplets to grow. It will also be important to confirm whether seipin works with other proteins as part of a molecular machine and, if so, to investigate how these proteins affect the formation and growth of lipid droplets.

## Introduction

Lipid droplets (LDs) are cellular organelles that act as reservoirs to store neutral lipids for metabolic energy and cell membrane components (for reviews, see *Gross and Silver, 2014*; *Hashemi and Goodman, 2015*; *Pol et al., 2014*; *Walther and Farese, 2012*; *Welte, 2015*; *Wilfling et al., 2014a*). Although most eukaryotic cells make LDs, the mechanism underlying the initial formation of LDs is mostly unknown. Excess storage of neutral lipids, such as triacylglycerols (TGs) and sterol esters, in LDs underlies many common metabolic diseases, such as obesity. Additionally, there is considerable interest in increasing cellular TG storage for industrial applications.

In essence, LD formation corresponds to establishing an oil-in-water emulsion (*Thiam et al., 2013b*). In cells, this appears to be a well-organized process and can be conceptually separated into several distinct biochemical steps that are particularly evident when cells are incubated with exogenous fatty acids to induce them to form TG-containing LDs. In the initial step, enzymes utilize fatty acids and glycerolipids to synthesize neutral lipids, such as TGs, in the endoplasmic reticulum (ER). TGs are subsequently packaged into nascent LDs that grow and are thought to bud from the ER to form initial LDs (iLDs). How cells utilize proteins to harness the principles of emulsion physics to control these initial steps in LD formation is unclear.

Much later, a second LD pathway responds to fatty acid loading of cells. A subset of the mature iLDs is converted to expanding LDs (eLDs) when specific TG synthesis enzymes [e.g., glycerol-3-phosphate acyltransferase (GPAT4) and acyl CoA:diacylglycerol *O*-acyltransferase 2 (DGAT2)] migrate from the ER to LDs via membrane bridges (*Wilfling et al., 2013*). Targeting of GPAT4 to LDs and formation of these bridges depends on the Arf1/COP-I proteins (*Wilfling et al., 2014b*). Another enzyme, CTP:phosphocholine cytidylyltransferase 1 (CCT1), binds to phosphatidylcholine (PC)-poor surfaces of eLDs, where it becomes activated and catalyzes the synthesis of PC to coat eLD surfaces (*Krahmer et al., 2011*). In this manner, eLDs grow dramatically through coordinated synthesis of core and surface lipids.

The current study focused on the initial steps in LD formation and on the ER-localized protein seipin, which has been implicated in LD formation with an uncertain role (for reviews, see *Cartwright and Goodman, 2012*; *Fei et al., 2011a*). The seipin gene was initially identified by mutations in rare but severe forms of congenital generalized lipodystrophy (*Magré et al., 2001*). Subsequently, LD morphology screens in *Saccharomyces cerevisiae* showed that the seipin homologue Fld1 is required for normal LDs; in its absence, cells have many small LDs or a few 'supersized' or giant LDs, depending on growth conditions (*Fei et al., 2008*; *Szymanski et al., 2007*). Seipin is an integral membrane protein with two transmembrane domains and a large, evolutionarily conserved ER luminal loop (*Agarwal and Garg, 2004*; *Lundin et al., 2006*). Seipin forms oligomers (*Binns et al., 2010*; *Sim et al., 2013*). In yeast, seipin localizes to ER-LD contact regions (*Grippa et al., 2015*; *Szymanski et al., 2007*; *Wang et al., 2014*), and yeast cells lacking seipin have abnormal LD formation (*Cartwright et al., 2015*; *Grippa et al., 2015*; *Wang et al., 2014*), suggesting a role for seipin in organizing this process. Alternatively, seipin might affect LDs by regulating lipid metabolism (*Boutet et al., 2009*; *Fei et al., 2011b*, *2008*, *2011c*; *Sim et al., 2012*; *Szymanski et al., 2007*; *Tian et al., 2011*; *Wolinski et al., 2015*) or by causing defects in ER calcium homeostasis (*Bi et al., 2014*).

Here, we investigated seipin function in LD formation in *Drosophila* and mammalian cells. We found that seipin acts at a distinct step of LD biogenesis, after nascent LDs form during iLD formation. Our data suggest that seipin localizes to ER-LD contact sites and enables nascent LDs to acquire more lipids from the ER and grow to form mature iLDs. Without seipin, this process appears to be blocked, resulting in massive accumulation of small nascent LDs. The few LDs that do grow exhibit aberrant targeting of lipid synthesis enzymes, such as GPAT4, involved in forming eLDs. The latter process likely explains the giant LD phenotype characteristically found in seipin-deficient cells.

## Results

### Seipin deficiency leads to altered LD morphology without evidence for altered lipid metabolism

As reported (*Fei et al., 2011b*, *2008*; *Szymanski et al., 2007*; *Tian et al., 2011*), we showed that depletion of seipin from *Drosophila* S2 cells by RNAi (~80% knockdown efficiency, *Figure 1—figure supplement 1A*) led to formation of giant LDs after prolonged oleic acid treatment to induce LD formation (*Figure 1A,* 24 hr). To determine the molecular basis of this phenotype, we examined when LD formation first appeared to be abnormal in seipin-deficient cells. Within 1 hr of adding oleic acid to cells, LDs in seipin-depleted cells were larger than those in control cells, although almost all LDs were less than 2 μm in diameter (*Figure 1A and B*, top). Giant LDs (diameter $\geq$ 2 μm) first appeared in seipin knockdown cells ~5 hr after adding oleic acid and were more prevalent after 8 hr. In contrast, giant LDs were rare in control cells. Seipin-depleted cells also had fewer LDs than control cells, particularly at later times (*Figure 1B*, bottom). Since the total areas with BODIPY-stained LD signal in optical sections of seipin-depleted cells and control cells at late time points were similar, the LDs likely coalesced in seipin-deficient cells.

The altered LD morphology during formation in seipin-deficient cells could result from changes in lipid synthesis, as suggested by some studies (*Boutet et al., 2009*; *Fei et al., 2011b*, *2008*, *2011c*; *Tian et al., 2011*). To examine this possibility, we used [$^{14}$C]-oleic acid as a tracer to measure lipid synthesis in seipin-depleted cells. Rates of accumulation of TG, PC, and phosphatidylethanolamine (PE) were similar in control and seipin knockdown cells both in cell homogenates (*Figure 1C*) and microsomes (*Figure 1—figure supplement 1B*), indicating similar rates of glycerolipid synthesis. Steady-state levels and synthesis rates of lipids in seipin-depleted cells showed no differences by high-resolution shotgun lipidomics (*Almeida et al., 2015*; *Ejsing et al., 2009*) at 3 hr after adding [$^{13}$C$_5$]-oleic acid (*Figure 1D*). To ensure the lack of differences in lipid synthesis was not due to residual seipin, we deleted seipin in human mammary carcinoma cells (SUM159) by CRISPR/Cas9-mediated genome editing (*Ran et al., 2013*) (*Figure 1—figure supplement 1C*). In this knockout clone, no seipin was detected (*Figure 1—figure supplement 1D*). LC-MS/MS lipidomics did not show evidence of altered lipid metabolism (for instance, levels of PC, PE or TG) between control and seipin knockout cells without oleic acid (*Figure 1—figure supplement 1E*).

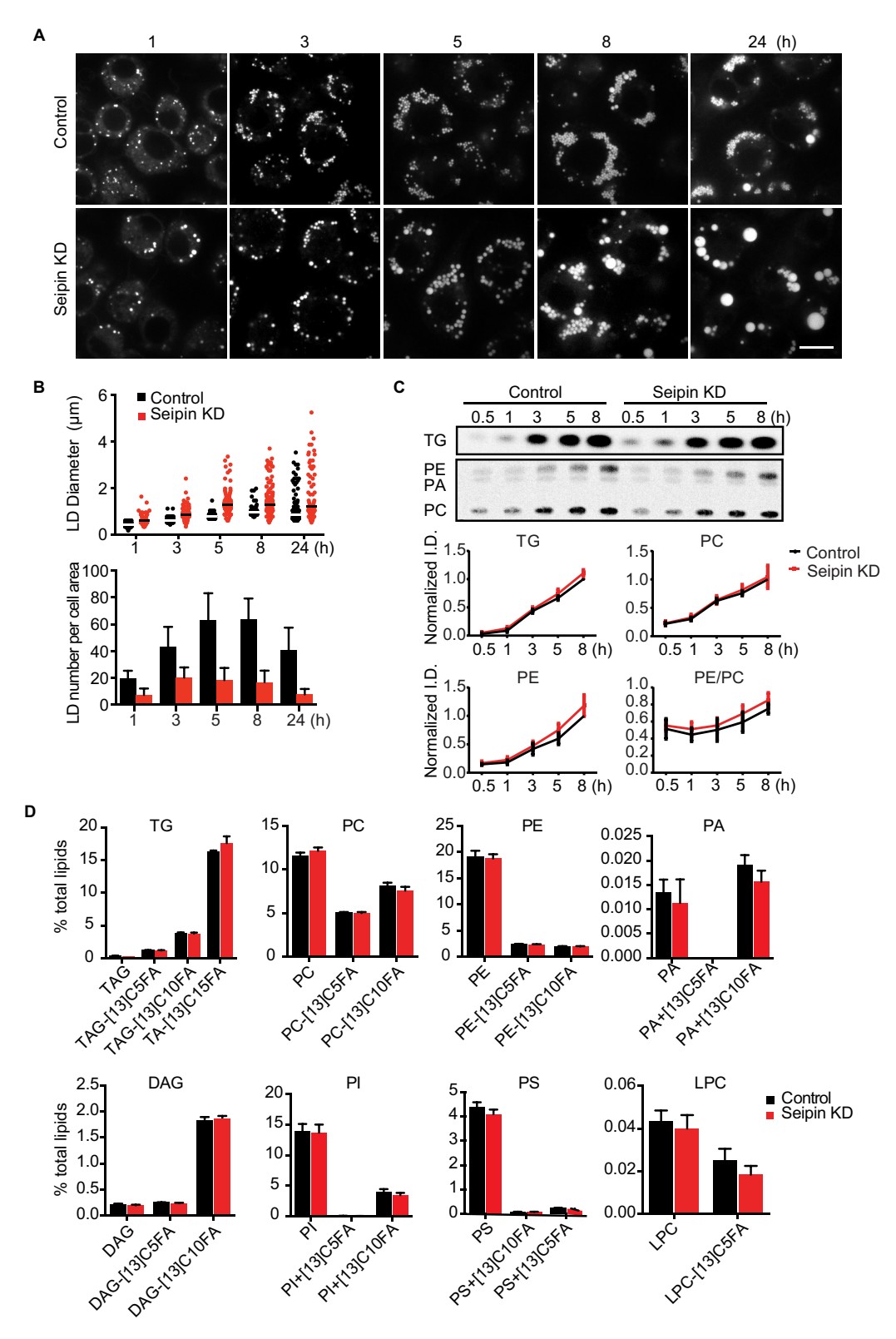

**Figure 1.** Seipin depletion alters LD morphology without affecting cellular lipid synthesis or composition in *Drosophila* S2 cells. (**A**) Time course of LD formation in control and seipin knockdown (KD) cells. S2 cells were treated with 1 mM oleic acid for the indicated times, and LDs were stained with BODIPY 493/503. Bar, 10 μm. (**B**) Quantification of LD formation over time. Top, LD diameters; lines show median. Bottom, average LD numbers per cell area. n = 20. (**C**) Seipin deficiency does not affect cellular glycerolipid synthesis. Cells were pulse-labeled with [14C]-oleic acid (100 μCi/μmol) for

*Figure 1 continued on next page*

*Figure 1 continued*

indicated times. Phospholipids and neutral lipids were extracted and separated by TLC. The TLC plate was exposed on an imaging screen, and the intensities of bands were quantified with FIJI software. Values are presented as integrated density normalized to protein concentration. n=3. (D) Seipin does not affect the flux and steady-state levels of lipids by lipidomics. Cells were labeled with [$^{13}C_5$]-oleic acid for 3 hr. Lipids were extracted, and lipid classes and species were identified by shotgun mass spectrometry–based lipidomics. n=3 biological replicates and 2 technical replicates.

The following figure supplements are available for figure 1:

**Figure supplement 1.** Seipin does not affect cell lipid synthesis or composition.

**Figure supplement 2.** Seipin knockdown does not affect ER morphology or stress.

Although others reported an accumulation of phosphatidic acid (PA) in seipin-deficient yeast (*Fei et al., 2011b*, *2011c*; *Sim et al., 2012*; *Tian et al., 2011*; *Wolinski et al., 2015*), we found no increase in cellular PA levels (*Figure 1D*, *Figure 1—figure supplement 1E*). Additionally, expression of a probe, GFP-PASS (*Lu et al., 2016*), that senses the accumulation of PA and possibly other anionic phospholipids (*Horchani et al., 2014*) did not accumulate at any specific site in cells during LD formation (*Figure 1—figure supplement 1F*). Since seipin resides in the ER membrane, changes in lipid composition might occur in the ER but be masked in global analyses by lipids from other membranes. However, the ER morphology appeared normal in seipin-depleted cells (*Figure 1—figure supplement 2A*), and we did not find evidence of altered ER lipid composition (*Figure 1—figure supplement 1B,E*). Thus, the abnormal LD phenotype found in seipin depletion does not likely result from defects in whole-cell or global ER lipid metabolism.

## Seipin deficiency results in aberrant accumulation of nascent LDs in contact with the ER

An alternative model of seipin's function posits a direct role in organizing TG for LD formation (*Cartwright and Goodman, 2012*). LDs are thought to form in several steps. First, TG is synthesized in the ER by lipid synthesis enzymes, such as DGAT1 (*Yen et al., 2008*). When the TG reaches a critical mass, it is thought to undergo phase separation and bud from the ER as iLDs. We hypothesized that seipin might act at one of these steps to facilitate iLD formation.

Since LDs were abnormal by 1 hr of formation in seipin knockdown cells, we examined earlier

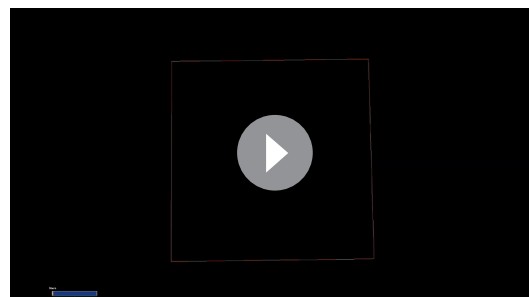

**Video 2.** Use of *LiveDrop* to visualize LD formation with lattice light-sheet microscopy in *Drosophila* S2 cells (related to *Figure 2A*). (A) A control cell expressing cherry-*LiveDrop* were seeded onto 5 mm coverslips, and entire cell volume of cells were imaged with custom-made lattice light-sheet microscopy at 4 s intervals and with the light-sheet and objective scan step size of 200 nm. Raw images were deconvolved and 3-D visualization was done with Amira software. The beginning and end sections of the movie (cyan) show the light-sheet and objective scanning through slices of the cell. The middle section of the movie (yellow) presents the 3-D reconstitution of the cell volume.

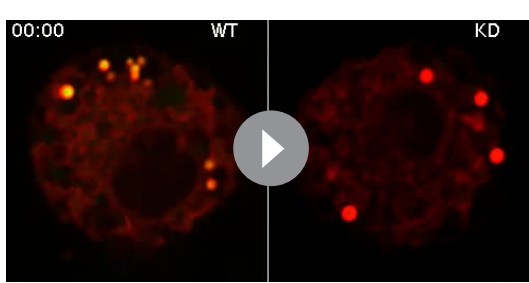

**Video 1.** Seipin depletion leads to aberrant accumulation of BODIPY-negative, *LiveDrop* in *Drosophila* S2 cells (related to *Figure 2A*). Cells expressing cherry-*LiveDrop* were treated with oleic acid and imaged with spinning disk confocal microscopy as described in *Figure 2A*. Green, BODIPY; red, cherry-*LiveDrop*. Time is presented as min: sec.

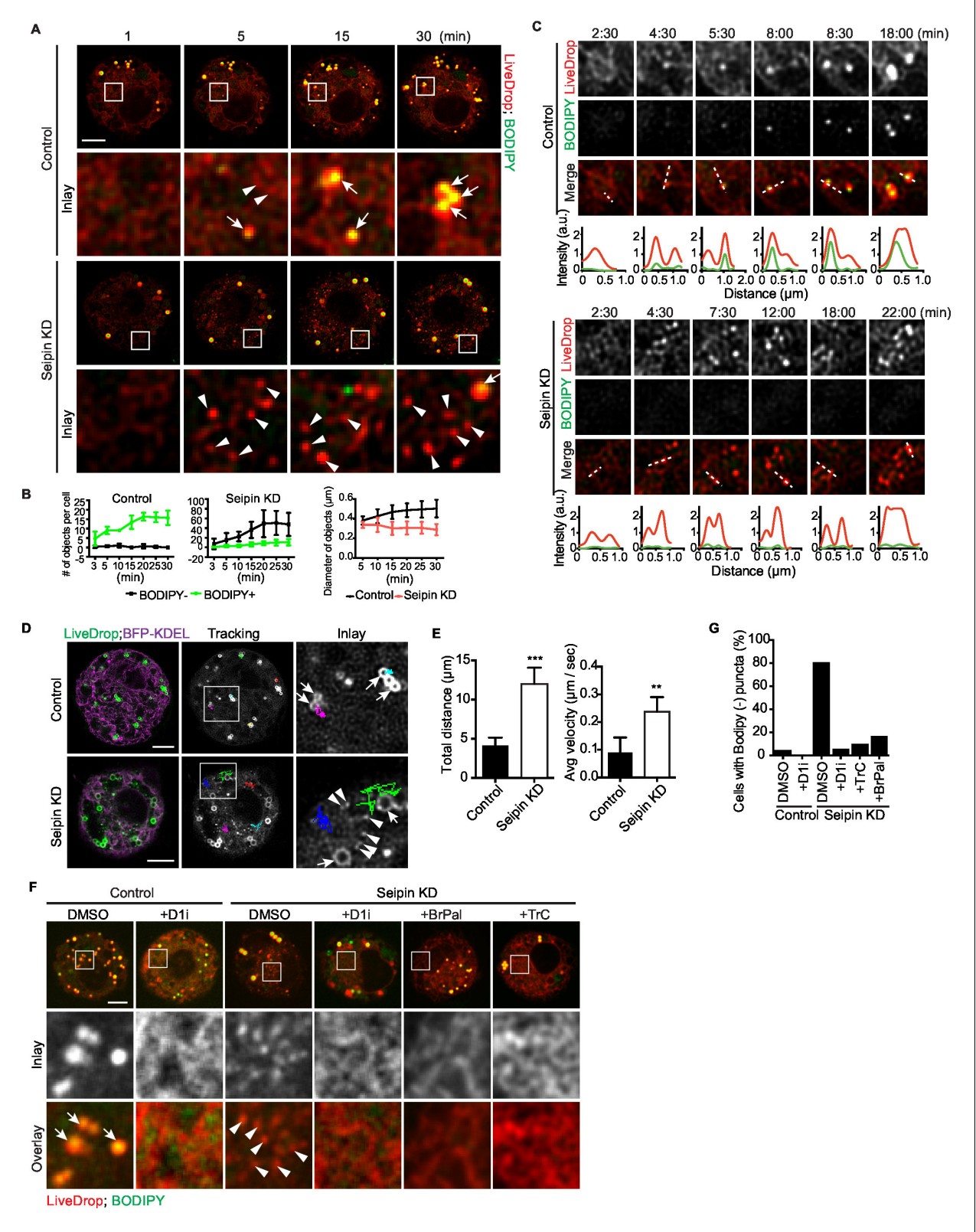

**Figure 2.** Aberrant accumulation of BODIPY-negative *LiveDrop* puncta in seipin-depleted *Drosophila* S2 cells. (A) Control or seipin knockdown cells expressing cherry-*LiveDrop* were treated with oleic acid immediately before the movie was taken at 30 s intervals for 30 min. Images were deconvolved as described in Materials and methods. Frames at indicated time points are shown. Green, BODIPY; red, cherry-*LiveDrop*. Arrow, BODIPY positive LDs; arrowhead, BODIPY negative puncta. Bar, 5 μm. (B) Quantification of numbers and size of *LiveDrop* objects that are positive or negative for BODIPY

*Figure 2 continued on next page*

*Figure 2 continued*

from movies taken at single optical plane. n=4. (C) Accumulation of *LiveDrop* puncta and BODIPY over time. Graphs show line profiles for each channel at indicated lines. Objects accumulate BODIPY overtime in control but not seipin knockdown cells. (D) *LiveDrop* puncta in the absence of seipin are highly mobile. Cells expressing GFP-*LiveDrop* and BFP-KDEL were incubated with oleic acid for 30 min, before live-cell images were taken at max speed (~0.38 s/frame) for 1 min. Movement of *LiveDrop* puncta are tracked with FIJI software. Representative tracks are shown. Bars, 5 µm. (E) Speed and distance of *LiveDrop* puncta movement were measured with FIJI. n= 3 cells, 8 puncta per cell. **p<0.005; ***p<0.001. (F) Presence of *LiveDrop* puncta in seipin-knockdown cells depends on TG synthesis. Cells expressing cherry-*LiveDrop* were treated with oleic acid for 30 min in the presence or absence of various TG synthesis inhibitors. D1i: DGAT1 inhibitor; BrPal: bromopalmitate; TrC: triacin C. Bar, 5 µm. Quantification of cells with abnormal accumulations of BODIPY negative, *LiveDrop* puncta are shown in (G). Representative results from two independent experiments are shown. 40 cells from each condition were quantified.

The following figure supplements are available for figure 2:

**Figure supplement 1.** Characterization of *LiveDrop* as an LD formation marker.

**Figure supplement 2.** *LiveDrop* is present in the LD fraction.

stages. Such studies require a fluorescent probe sensitive enough to detect TG collections during the earliest steps of LD formation. These normally have too little lipid mass to be detected by conventional neutral lipid dyes (e.g., BODIPY) that partition preferentially into the neutral lipid phase. We therefore developed a probe, which we named *LiveDrop*, that contains the membrane hairpin domain (amino acids 160–216) of the glycerolipid synthesis enzyme GPAT4 fused to a fluorescent protein (*mCherry or eGFP*) (*Wilfling et al., 2013*) (*Figure 2—figure supplement 1A*). Although full-length GPAT4 normally localizes only to eLDs (*Wilfling et al., 2013*), *LiveDrop* localizes from the ER to all newly synthesized LDs upon their formation (*Figure 2A,C*, control) and identifies TG collections before BODIPY (as seen in *Figure 2C*). *LiveDrop* localization to foci that become LDs depends on TG synthesis (*Figure 2F* and *Figure 2—figure supplement 1D*), and in in vitro experiments using adhesive emulsion (*Thiam et al., 2012*), we found that *LiveDrop* partitions preferentially into mono-layers over bilayer membranes compared with a control protein Arf1 (*Figure 2—figure supplement 1C*). Together, these data demonstrate that *LiveDrop* is a sensitive probe, identifying the earliest steps of LD formation. The finding that LiveDrop localizes to all newly synthesized LDs whereas the full-length GPAT4 localizes only to eLDs suggests a retention mechanism that maintains GPAT4 in the ER during LD formation.

We used *LiveDrop* and live-cell imaging to study LD formation in *Drosophila* S2 cells. In control cells, when LD formation was induced by oleic acid, *LiveDrop*, which was initially diffuse within the ER, rapidly accumulated in small puncta that grew larger within minutes and subsequently became detectable with BODIPY (*Figure 2A,B,C*,

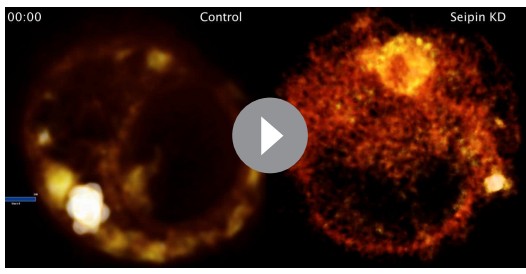

**Video 3.** Visualization of LD formation with lattice light-sheet microscopy in control and seipin knockdown *Drosophila* S2 cells (related to *Figure 2A*). Control or seipin knockdown cells expressing cherry-*LiveDrop* were imaged and processed as above. Video shows the 3-D projection of each cell section containing 1/3 of the cells thickness. Time is presented as min: sec.

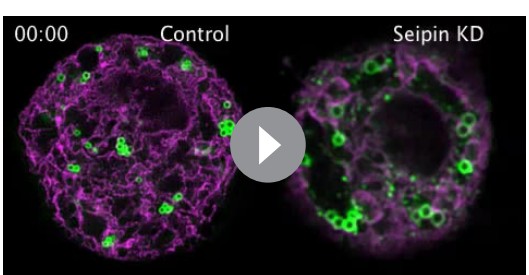

**Video 4.** *LiveDrop* punta in seipin-depleted cells are highly mobile (related to *Figure 2D*). *Drosophila* S2 cells expressing GFP-*LiveDrop* and BFP-KDEL were treated with oleic acid and imaged and deconvolved as described in *Figure 2D*. Time is presented as sec: msec.

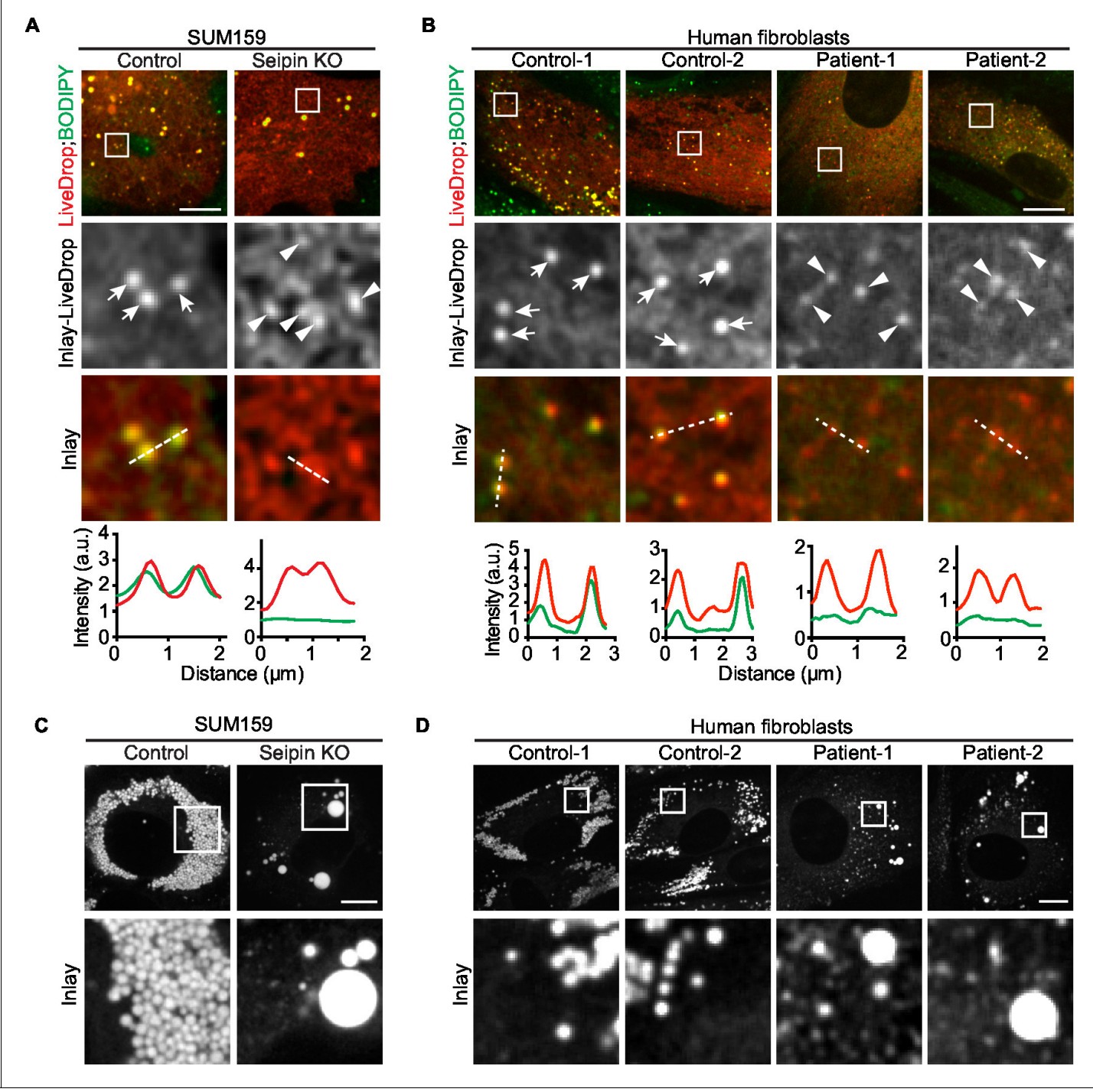

**Figure 3.** Seipin's functions in initial LD formation and late LD phenotype are evolutionarily conserved. (**A**) *LiveDrop* puncta accumulate in SUM159 mammary carcinoma cells lacking seipin. Cells were incubated with oleic acid for 30 min before imaging. Red, cherry-*LiveDrop*; green, BODIPY. Arrows, BODIPY positive LDs; arrowheads, BODIPY negative *LiveDrop* puncta. Bar, 5 μm. Graphs show the line profile for each channel at dotted lines. (**B**) *LiveDrop* puncta accumulate in primary human fibroblasts from two subjects with lipodystrophy due to *BSCL2* loss-of-function mutations (patient 1, *p. A212fsX231*; patient 2, *p.T109Nfs*5* & *p.P65Gfs*28*). Bar, 5 μm. (**C**) LD phenotype in seipin knockout SUM159 cells at late stage of formation. Cells were treated with oleic aicd for 16 hr before imaging. Bar, 5 μm. (**D**) LD phenotype during late stage of formation in primary fibroblasts from healthy controls or lipodystrophy patients. Bar, 10 μm.

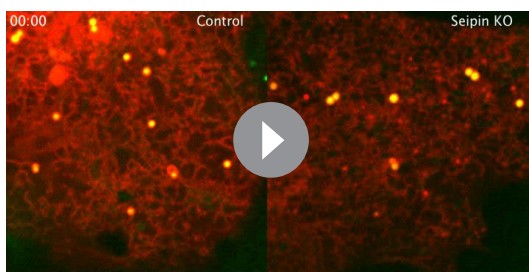
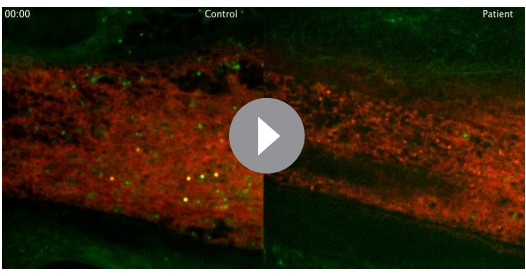

**Video 5.** Accumulation of *LiveDrop* puncta in seipin knockout SUM159 cells (related to *Figure 3A*). Wildtype or seipin knockout SUM159 cells expressing cherry-*LiveDrop* were incubated with oleic acid immediately before imaging with spinning disk confocal microscopy. Images were taken at 10 s intervals for 30 min. Red, cherry-*LiveDrop*; green, BODIPY.

**Video 6.** Accumulation of *LiveDrop* puncta in human fibroblasts from a lipodystrophy patient with loss of function mutations in seipin (related to *Figure 3B*). Fibroblasts from an apparently healthy control subject and a lipodystrophy patient with seipin mutations (*p. T109Nfs*5* & *p.P65Gfs*28*) were transfected with cherry-*LiveDrop* and incubated with oleic acid immediately before imaging with spinning disk confocal microscopy. Images were taken at 10 s intervals for 30 min. Red, cherry-*LiveDrop*; green, BODIPY.

*Figure 2—figure supplement 1B*, and *Videos 1*, *2*). In contrast, the ER of seipin-deficient cells had large numbers (up to an average of 60 per cell, *Figure 2B*) of small *LiveDrop* puncta that mostly did not grow large enough to stain with BODIPY (*Figure 2A,B,C*, *Figure 2—figure supplement 1B*, and *Videos 1*, *3*). Instead, they stayed small and highly mobile (0.23 ± 0.06 μm/s) at the ER (*Figure 2D,E* and *Video 4*). However, a few larger, BODIPY-positive LDs formed in many of the seipin-depleted cells, possibly due to TG phase separation in a non-seipin organized process. BODIPY-stained LDs formed in control cells were relatively static (0.07 ± 0.02 μm/s), compared with LiveDrop puncta in seipin-depleted cells (*Figure 2D,E* and *Video 4*). To clarify the identity of *LiveDrop* puncta in seipin-knockdown cells, we inhibited TG synthesis with pharmacological inhibitors (*Figure 2F,G*) or by knocking down TG synthesis enzymes (*Figure 2—figure supplement 1D,E*). These treatments abolished the *LiveDrop* puncta, indicating that they depend on TG synthesis and are likely very small TG collections associated with the ER.

## Seipin function in iLD formation is evolutionarily conserved

To assess evolutionary conservation of seipin function, we examined two mammalian cell models with seipin deficiency. In addition to the SUM159 seipin knockout cell line, we also studied primary fibroblasts from controls and patients with congenital generalized lipodystrophy type 2, in which frame-shift mutations in *BSCL2* (mammalian seipin) [patient 1: *p.A212fsX231*; patient 2: *p.T109Nfs*5* (*Agarwal et al., 2003*) & *p.P65Gfs*28*] co-segregate with the disease phenotype. In each mammalian seipin-defective cell line, *LiveDrop* formed numerous, BODIPY-negative puncta upon oleate treatment (*Figure 3A,B*, and *Videos 5*, *6*), recapitulating the findings from seipin knockdown S2 cells. After prolonged oleic acid treatment (24 hr), giant LDs formed in each seipin deficiency model. However, unlike *Drosophila* S2 cells, there were very few giant LDs and more numerous small puncta that were weakly detected by BODIPY staining (*Figure 3C,D*). Of note, ER morphology seemed normal in seipin-knockout cells (*Figure 1—figure supplement 2A*), and there were no changes in ER lipid composition (*Figure 1—figure supplement 1E*) or in the activity of ER stress pathways (*Figure 1—figure supplement 2B,C*).

Across species, seipin contains highly conserved residues in the two transmembrane domains and the ER luminal loop, but the short N- and C-terminal cytosolic regions are not conserved. To test whether the conserved regions are important for its function in iLD formation, we overexpressed fluorescently tagged regions of seipin in a setting of seipin depletion in the S2 cell model. We knocked down seipin with dsRNA targeting the 3′UTR to avoid degradation of expressed seipin proteins. These knockdowns were somewhat less efficient than targeting the open reading frame, with ~65% of cells exhibiting accumulation of BODIPY-negative *LiveDrop* puncta (*Figure 4A,B*). When full-length *Drosophila* seipin was expressed in the knockdown cells, the numbers of *LiveDrop* puncta

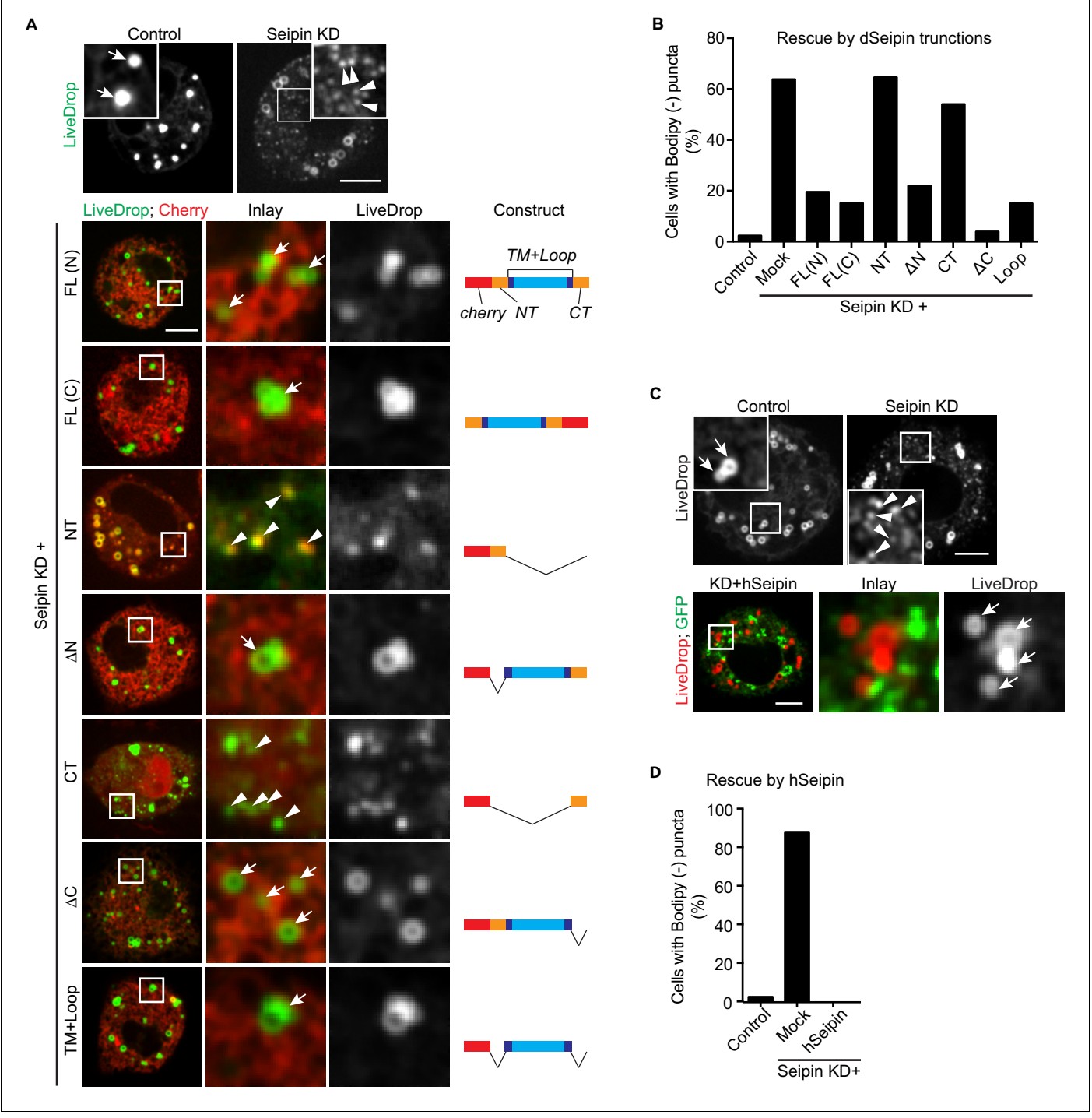

**Figure 4.** The conserved transmembrane domains and ER luminal loop of seipin are important for its function in initial LD formation in *Drosophila* S2 cells. (**A**) Prevention of the seipin-depletion phenotype by expression of the transmembrane domains and ER loop, but not by the N- or C-terminus of *Drosophila* seipin. Gene knockdowns were performed with control or seipin dsRNAs targeting 3'UTR region of seipin. Cells were then co-transfected with GFP-*LiveDrop* (green) and cherry-tagged *Drosophila* seipin truncation mutants (red). Seipin constructs used are illustrated on the right. Cells were treated with oleic acid for 30 min before imaging. Arrows, BODIPY-positive LDs; arrowheads, BODIPY-negative *LiveDrop* puncta. Bars, 5 μm. Quantification of cells with abnormal accumulations of BODIPY-negative *LiveDrop* puncta are shown in (**B**). Representative results from two independent experiments are shown. 40 cells from each condition were quantified. (**C**) Expression of human seipin (hSeipin) prevents the seipin-depletion phenotype. Gene knockdowns were performed with control or dsRNAs targeting coding region of seipin. Cells were then co-transfected with cherry-*LiveDrop* (red) and human seipin (green). Cells were treated with oleic acid for 30 min before imaging. Arrows, BODIPY-positive LDs;
*Figure 4 continued on next page*

*Figure 4 continued*

arrowheads, BODIPY-negative *LiveDrop* puncta. Bars, 5 µm. Quantification of cells with abnormal accumulations of BODIPY negative *LiveDrop* puncta are shown in (D). Representative results from two independent experiments are shown. 40 cells from each condition were quantified.

were dramatically reduced in most cells. Tagging with a fluorescent protein localized to the N- or C-terminus resulted in similar rescue efficiency 30 min after oleate loading (*Figure 4A,B*). Deletion of the N- or C-terminal cytosolic domain or both did not affect the rescue efficiency of seipin overexpression. In contrast, expression of the N- or C-terminus alone did not rescue the phenotype (*Figure 4A,B*). These results suggest that the evolutionarily conserved regions of seipin (i.e., the ER luminal domain and transmembrane domains) are required for seipin's function in organizing iLD formation. In agreement with evolutionary conservation of seipin's function, expression of human seipin in seipin-depleted *Drosophila* S2 cells completely abolished abnormal *LiveDrop* puncta accumulation, and mature iLDs formed normally in all transfected cells (*Figure 4C,D*).

## Electron-tomography reveals nascent LD intermediates accumulation in seipin-deficient cells

To visualize the molecular architecture of the *LiveDrop* puncta at nanometer resolution, we analyzed control and seipin-knockout SUM 159 cells by transmission electron microscopy. Consistent with images obtained by live cell imaging, wildtype cells contained numerous mature iLDs ranging in diameter between 200–500 nm, with an average of ~300 nm, after 60 min of oleate treatment (*Figure 5A,B*). In contrast, seipin-deficient cells showed a large accumulation of nascent LDs of uniform size of less than 200 nm diameter, with an average of ~170 nm (*Figure 5A,B*). These nascent LDs were most often located in close proximity to the ER. To determine whether the monolayer covering them was continuous with or separate from the ER membrane, we used electron tomography to examine thick sections of cells. We found that the nascent LDs in seipin-deficient cells were always in close proximity to the ER and that their surface monolayer was separated from the ER membrane (*Figure 5C,D* and *Videos 7*, *8*, *9* and *10*). Modeling of the tomograms indicated that ribosomes appeared to be absent from the space between the nascent LDs and the ER and that this space sometimes had electron density *Figure 5D* and *Video 11*. Taken together, these data suggest that the *LiveDrop* puncta represent nascent LDs that are separate from, but in close contact with the ER,

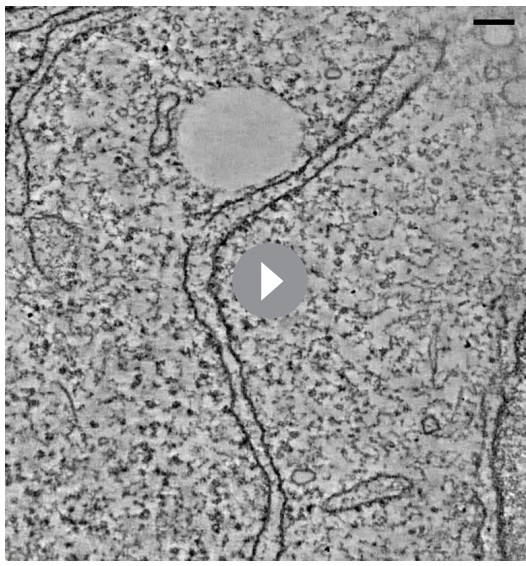

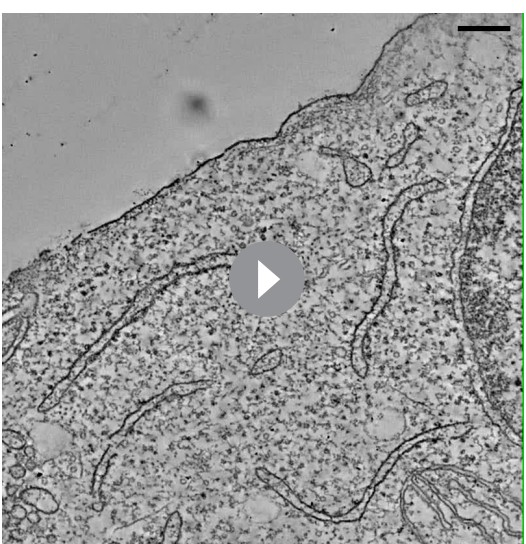

**Video 7.** Electron tomography of LDs in control and seipin knockout SUM159 cells with dual axis (related to *Figure 5C*). Bar, 100nm.

**Video 8.** Electron tomography of LDs in control and seipin knockout SUM159 cells with dual axis (related to *Figure 5C*). Bar, 200 nm.

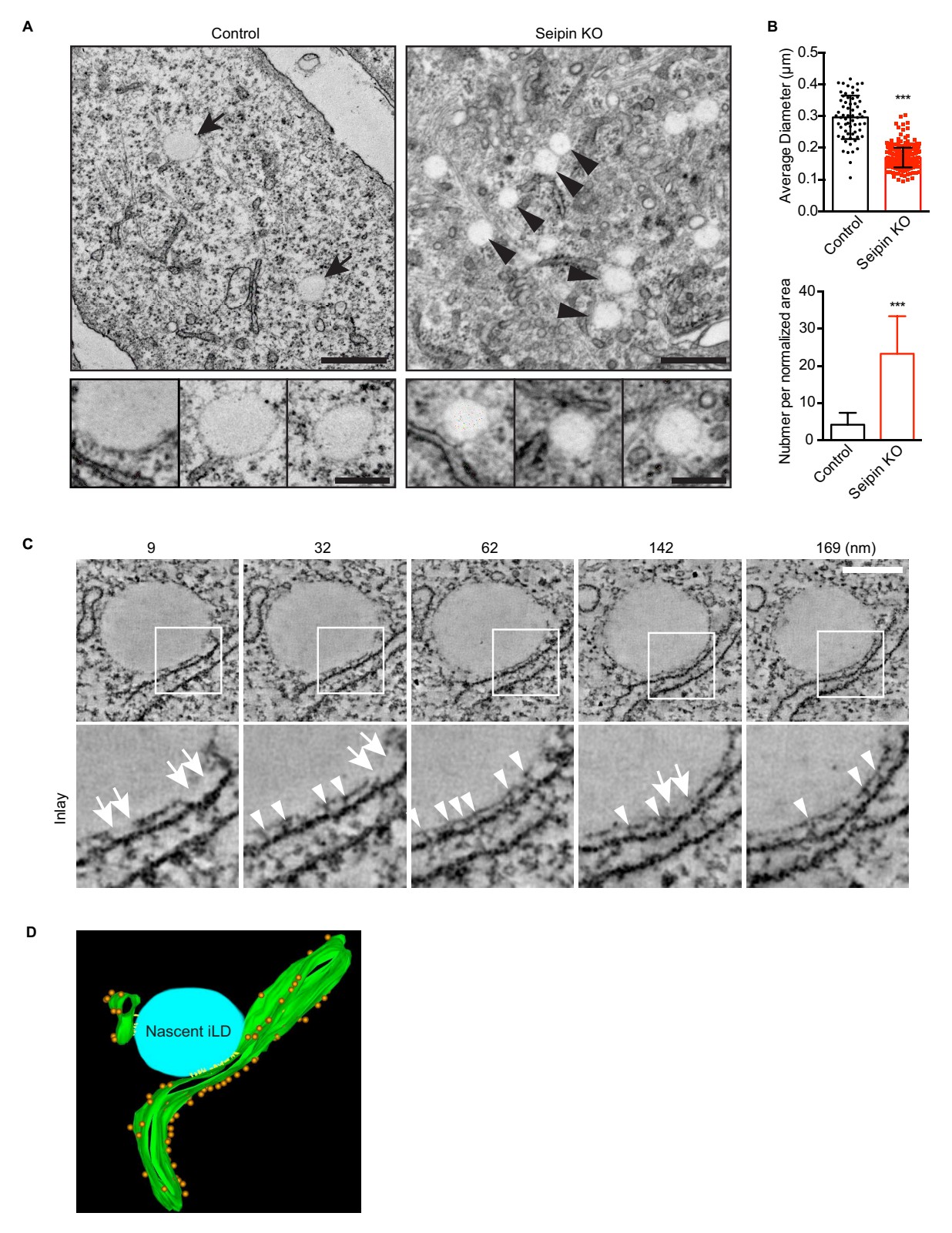

**Figure 5.** Seipin-deficient cells accumulate nascent LDs that are close to the ER. (**A**) Control and seipin knockout SUM159 cells were fixed, embedded, and stained with uranyl acetate and lead citrate. Thin sections were imaged with transmission electron microscopy. Arrows, mature iLDs; arrowheads, nascent LDs. Close-up images (bottom row) are from different cell areas, representing different levels of proximity to the ER. Bars, 0.5 μm (top) and 0.2 μm (bottom). (**B**) Quantification of number and size of LDs in control and seipin-knockout SUM159 cells. Total of 15 areas from each phenotype were

*Figure 5 continued on next page*

*Figure 5 continued*

quantified. ***p<0.0001. (**C**) Electron tomography of a nascent LD in the seipin-knockout SUM159 cell. Thick sections of cells were imaged and reconstituted with IMOD. Serial sections from one example are shown. Numbers on top represent the relative position. Arrows, membrane contact zone; arrowheads, possible filamentous structures between nascent LDs and the ER. Bar, 0.2 μm. (**D**) Modeling of electron tomograms. Green, ER; blue, nascent LDs; orange, ribosomes; yellow, possible filamentous structure between nascent LDs and the ER. Rendering was performed with IMOD software. Movies of the tomograms are shown in *Video 11*.

The following figure supplement is available for figure 5:

**Figure supplement 1.** Seipin-deficient cells accumulate small, nascent LDs that are close to the ER.

and that fail to grow to the size of normal, mature iLDs.

## Seipin forms dynamic foci within the ER that interact with nascent LDs and promote LD growth

Because our data suggested a role for seipin in the growth of nascent LDs to mature iLDs, we used live imaging to visualize the protein during LD formation. We fluorescently tagged the N-terminus of seipin with GFP at its endogenous locus in S2 cells by CRISPR/Cas9-mediated genome editing (*Housden et al., 2015*). Without oleic acid, GFP-seipin formed distinct foci in cells that overlapped with an ER marker (ss-BFP-KDEL; (*Dayel et al., 1999*) (*Figure 6A*). These foci did not localize to any specific domain of the ER (e.g., tubules, sheets, tips) (*Figure 6A* and *Figure 6—figure supplement 1A*) or three-way junctions marked by lunapark (*Figure 6—figure supplement 1B*), and they moved rapidly along the ER (1.06 ± 0.22 μm/s; *Figure 6B*, *Figure 6—figure supplement 1A* and *Video 12*). Upon adding oleic acid to form LDs, *LiveDrop* puncta initially were distinct from GFP-seipin foci. However, by 5 min, many *LiveDrop* puncta and seipin foci became co-localized. Once this occurred, the individual *LiveDrop* punctum was stabilized and began to grow in volume (*Figure 3C* and *Video 13*).

These data suggest a model in which, rather than fixed structures, seipin foci are highly mobile and survey the ER for nascent LDs. When they encounter one, as detected by *LiveDrop*, they interact with the nascent LD to enable its growth to a mature iLD. Consistent with this model, seipin foci associated with nascent LDs became relatively immobile (0.52 ± 0.09 μm/s), whereas those not associated with nascent LDs were highly dynamic (0.89 ± 0.19 μm/s; *Figure 6D*).

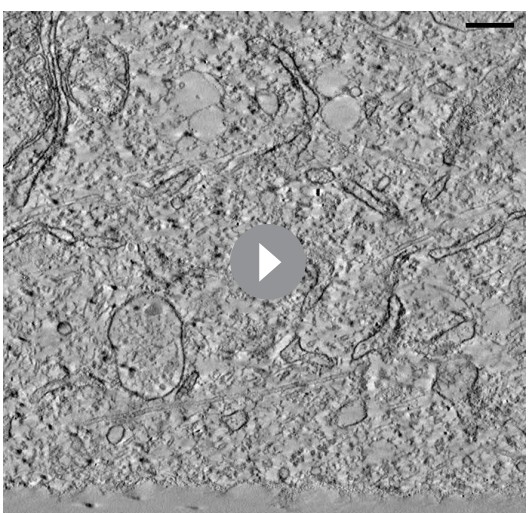

**Video 9.** Electron tomography of LDs in control and seipin knockout SUM159 cells with single axis (related to *Figure 5C*). Bar, 200 nm.

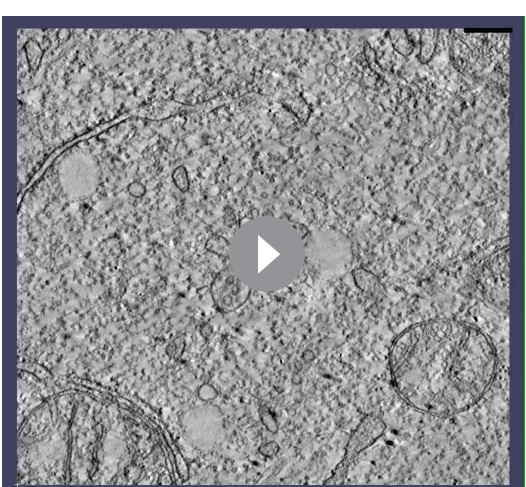

**Video 10.** Electron tomography of LDs in control and seipin knockout SUM159 cells with single axis (related to *Figure 5C*). Bar, 200nm.

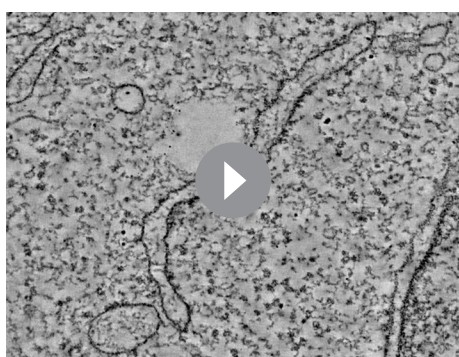

**Video 11.** Modeling of electron tomograms shown in *Video 7* (related to *Figure 5D*). Green, ER; blue, nascent LDs; orange, ribosomes; yellow, possible filamentous structure between nascent LDs and the ER. Rendering was performed with IMOD software.

At later times during LD formation, seipin remained in close proximity to LDs. At 3 hr or later after adding oleic acid, almost every LD was associated with at least one seipin focus (*Figure 6E*). However, there were many more seipin foci than LDs, and many seipin foci did not localize to LDs.

## Abnormal LD formation in seipin deficiency leads to more and earlier formation of expanding LDs

We showed that seipin deficiency leads to abnormal LDs (*Figure 1A,B*). It was unclear, however, how arrested growth early in LD formation leads to the phenotype of giant LDs in late stages of formation. Accumulation of nascent iLD intermediates in the early stages might indirectly result in fewer but larger LDs. We hypothesized that the LD expansion pathway, catalyzed by TG synthesis on eLDs, is aberrantly activated during seipin deficiency. In the LD expansion pathway, GPAT4 targets to selected LDs to initiate localized TG synthesis to expand LDs, and GPAT4 is thus a marker for eLDs (*Wilfling et al., 2013*). Immunofluorescence of endogenous GPAT4 in seipin-depleted cells showed that GPAT4 targets to LDs as early as 10 min after LD formation is induced by oleic acid. In control cells, GPAT4 localizes to LDs much later (*Figure 7A*) (not until 1 hr after induction of LDs, not shown). At 20 min after adding oleic acid, 18% of LDs in seipin knockdown cells were marked by GPAT4, but only 1–2% in control cells were GPAT4-positive (*Figure 7A,C*). After prolonged oleic acid treatment (8 hr), 8% of LDs in wild-type cells were GPAT4-positive, compared with 72% in seipin-depleted cells (*Figure 7B,C*). In fact, most of the GPAT4 pool was localized to LDs in seipin-depleted cells, and little remained in the ER (*Figure 7B*). This observation suggests that accumulation of nascent iLDs in seipin deficiency caused aberrant GPAT4 targeting to LDs, thereby initiating localized TG synthesis and LD expansion prematurely.

If the model of aberrant eLD formation in seipin deficiency is correct, generation of giant LDs in seipin deficiency should depend on GPAT4 and the LD expansion pathway. To test this prediction, we examined the effect of seipin knockdown, combined with knocking down TG synthesis enzymes in the LD expansion pathway (i.e., GPAT4, AGPAT3, DGAT2), and compared this with knockdowns of TG synthesis enzymes that do not mediate LD expansion (i.e., GPAT1, AGPAT2, DGAT1; *Wilfling et al., 2013*). Knockdown of enzymes of the LD expansion pathway abolished the giant LDs (diameter > 2.5 µm) of seipin deficiency, but knockdown of enzymes of the non-LD pathway did not affect LD size in seipin knockdown cells (*Figure 7D*, yellow box).

## Giant LDs of seipin-deficient cells are deficient in phospholipids

Rapid LD expansion and increased localized TG synthesis in seipin-deficient cells could lead to insufficient phospholipid surfactants on the surface monolayers of LDs to cover the hydrophobic cores. This, in turn, would lead to increased surface tension on these eLDs and likely to LD coalescence (*Krahmer et al., 2011*).

To test this prediction, we directly measured [$^{14}$C]-oleic acid incorporation into TG and phospholipids in LDs from control or seipin-depleted cells. Consistent with our earlier measurements, incorporation of radioactivity into total cellular PC and TG was similar in wild-type and seipin knockdown cells (not shown). However, incorporation of label into phospholipids (PC and PE) specifically on LDs was decreased in seipin-depleted cells at 5–8 hr after adding oleic acid to cells, indicating a relative deficiency of phospholipids in LDs of seipin-depleted cells at these late times. As a consequence, the ratio of surface phospholipids to TG (i.e., PC/TG or PE/TG) decreased markedly at this time (*Figure 8A*). This observation is consistent with the time course of LD formation (*Figure 1A*), with

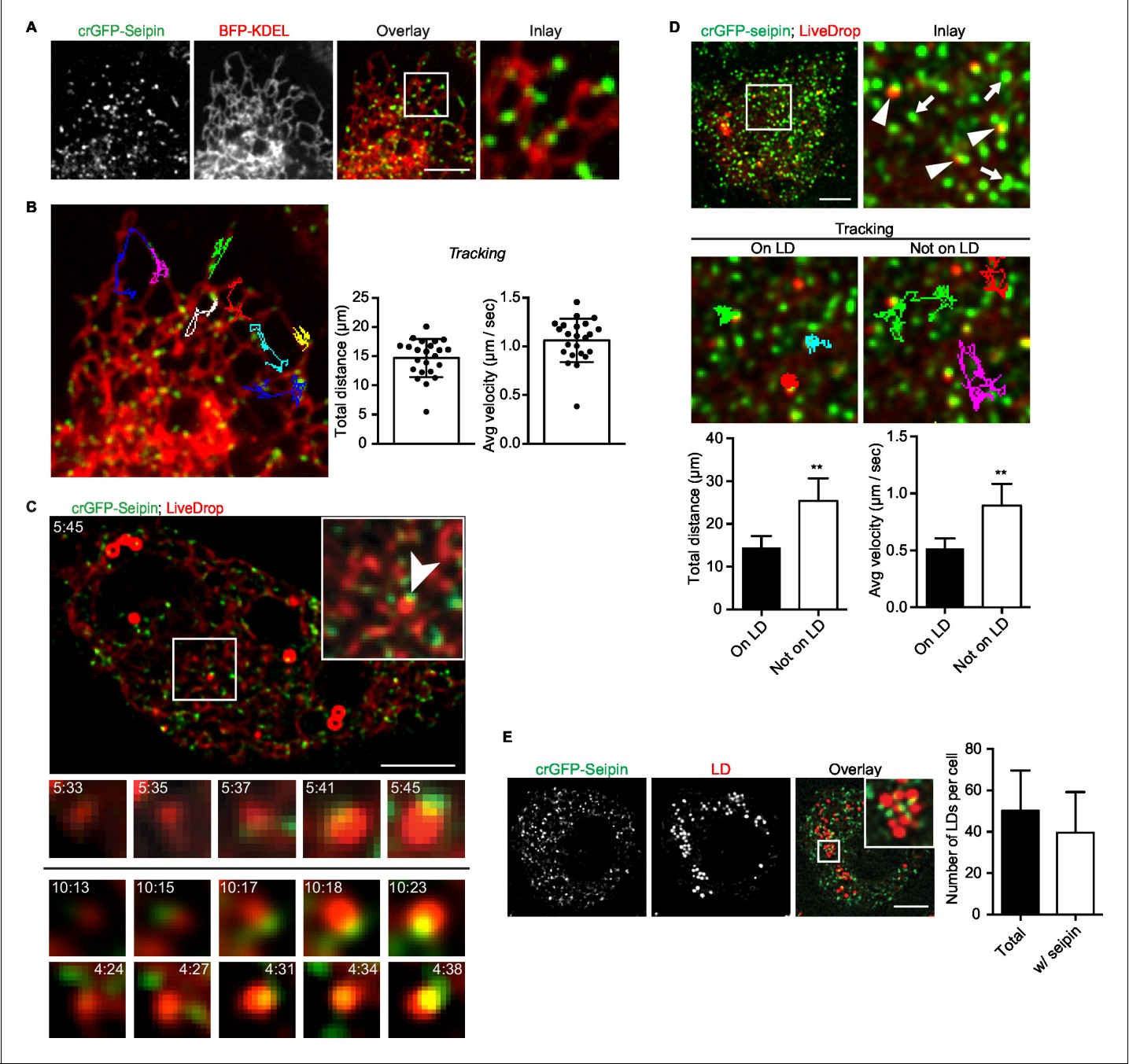

**Figure 6.** Endogenous seipin forms discrete, mobile foci in the ER that co-localize with initial LDs during formation in *Drosophila* S2 cells. (**A**) Endogenous seipin was tagged with GFP at the N-terminus (crGFP-seipin, green) by CRISPR/Cas9 genome editing. Cells were transfected with BFP-KDEL (red), and movies were captured with live-cell microscopy at 0.31 s frame intervals in the absence of oleic acid. The first frame is shown. Bar, 5 μm. (**B**) Seipin forms discrete foci that are highly mobile along the ER. Movement of GFP-seipin foci from movies taken in (**A**) was tracked with FIJI software. Total distance and average velocity were calculated. n= 3 cells, 8 foci per cell. (**C**) Seipin foci become co-localized with *LiveDrop* puncta to form nascent growing LDs. crGFP-seipin cells expressing cherry-*LiveDrop* were incubated with oleic acid for 5 min before live-cell images were taken (~2.05 s frame interval) for 2 min. Images were de-convolved as described in Methods. Top: image of a whole cell at indicated time point; inlay: magnification of boxed area; arrowhead: an event where a *LiveDrop* punctum becomes associated with seipin and grows. A time series of this event is magnified under the image. Bottom: two more examples of similar events in other cells. Green, GFP-seipin; red, cherry-*LiveDrop*. Bar, 5 μm. (**D**) Seipin foci associated with LDs become less mobile. crGFP-seipin cells expressing cherry-*LiveDrop* were treated with oleic acid for 30 min. Live-cell images were taken at max speed (~0.27 s frame interval) for 30 s, and the movement of seipin foci that are associated (arrowheads) or not associated (arrows) with LDs were traced and analyzed with FIJI. Representative tracks overlaid on image are shown (middle). Total distance and average velocity were calculated (bottom). n= 3 cells, 6 puncta per cell. Bar, 5 μm. **p<0.005. (**E**) The majority of formed LDs are associated with seipin foci. crGFP-seipin cells were treated with oleic

*Figure 6 continued on next page*

*Figure 6 continued*

acid for 3 hr, and LDs were stained with LipidTox (red). Images were taken as Z stacks, and middle slices are shown. Number of LDs with or without a seipin focus within a cell was quantified. Bar, 5 µm. n=10 cells.

The following figure supplement is available for figure 6:

**Figure supplement 1.** Seipin foci are localized in the ER but not at specific ER domains.

giant LDs forming 5–8 hr after adding oleic acid, and suggests phospholipid deficiency on LDs results in their coalescence to form giant LDs.

Our previous studies in *Drosophila* S2 cells established that PC-deficient LDs recruit CCT1, the rate-limiting enzyme of PC synthesis, to the surface monolayer, where the enzyme becomes activated for maintaining PC homeostasis (*Krahmer et al., 2011*). If phospholipids, specifically PC, are deficient on eLDs in seipin-deficient cells, CCT1 should have increased targeting to LDs during formation. Indeed, seipin-depleted cells exhibited earlier and markedly increased CCT1 targeting to LDs, compared with control cells (*Figure 8B*). Further, when seipin-deficient cells were treated with PC liposomes or choline supplementation, no giant LDs (diameter > 2.5 µm) in seipin deficiency were formed (*Figure 8C*, yellow box). In contrast, PC liposomes corrected the phenotype of CCT1 deficiency, and as expected, choline supplementation did not. We also altered phospholipid synthesis by knock down of CCT1 or CTP:phosphoethanolamine cytidylyltransferase (ECT) and tested the effect on seipin knockdown. ECT knockdown is predicted to shift the balance of phospholipid synthesis to PC from PE. As expected, CCT1 depletion aggravated, and ECT depletion abolished, the seipin knockdown phenotype (*Figure 8D*). These data suggest that faster LD expansion in seipin-depleted cells leads to more eLDs with transient PC deficiency, and this likely leads to LD coalescence to form giant LDs.

## Discussion

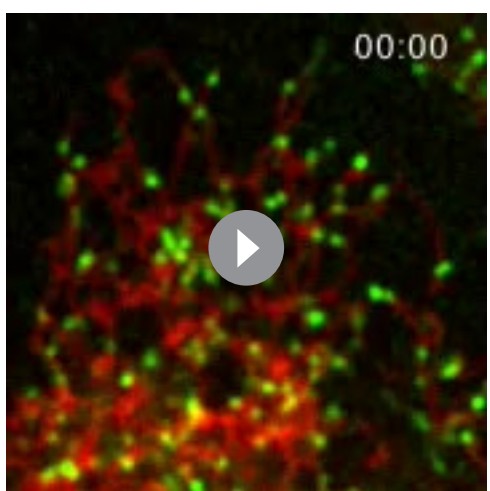

**Video 12.** Endogenously tagged GFP-seipin forms foci that move dynamically along the ER in S2 cells (related to *Figure 6B*). crGFP-seipin cells were prepared and imaged as described in *Figure 6A*. Time is presented as sec: msec. Seipin foci move very dynamically and do not localize to specific ER domain.

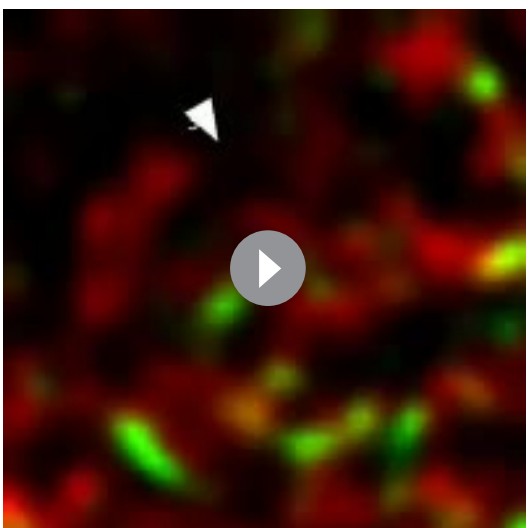

**Video 13.** Endogenously tagged GFP-seipin encounters and stabilizes *LiveDrop* puncta (related to *Figure 6C*). Cells were treated, imaged and deconvolved as described in *Figure 6C*. Frame rate: 2 s. Note that a *LiveDrop* punctum becomes stabilized and grows in volume after association with a seipin focus.

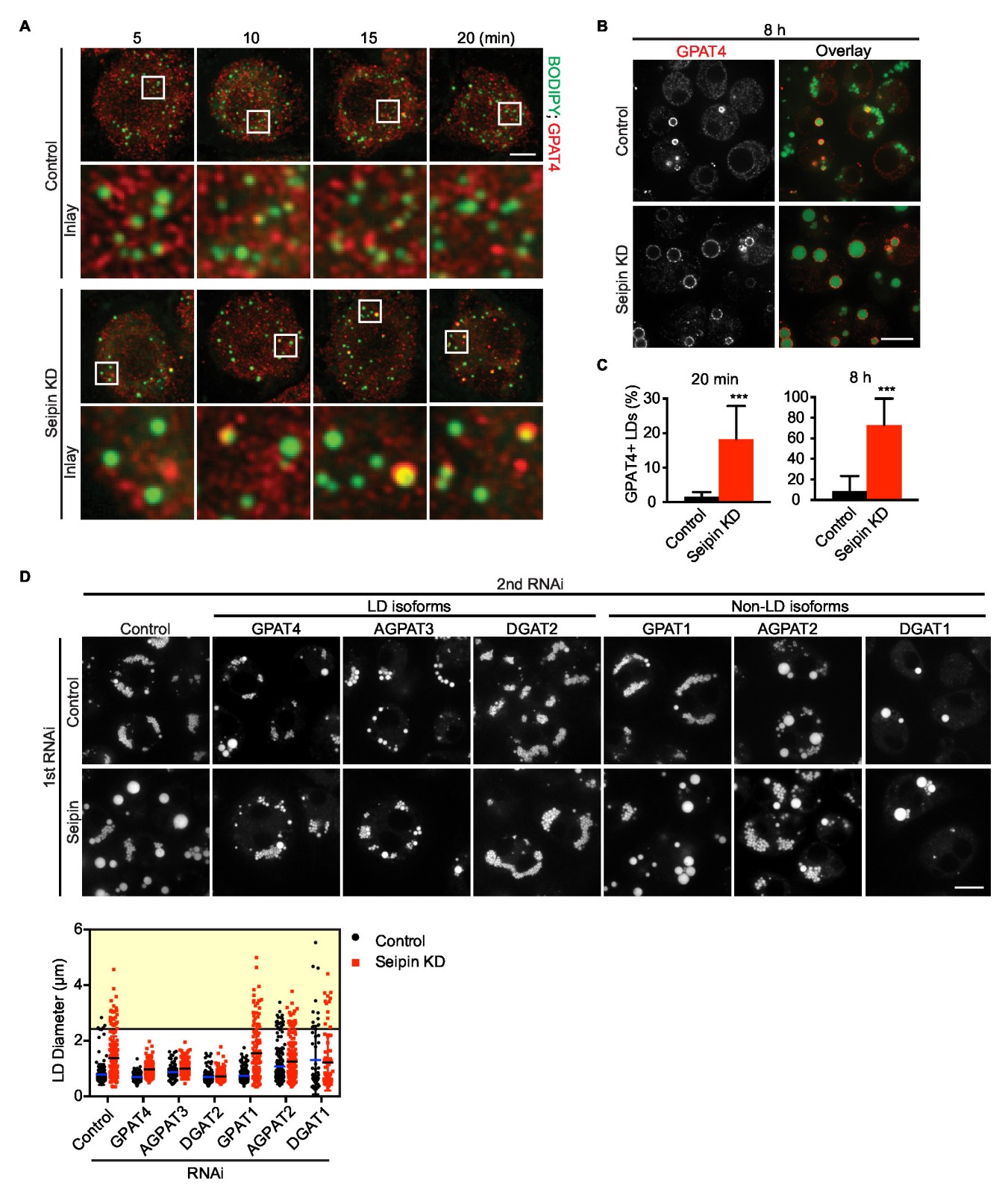

**Figure 7.** Aberrant LD formation in seipin deficiency increases the expanding LD population in *Drosophila* S2 cells. (**A**, **B**) Earlier and increased GPAT4 targeting to initial LDs with seipin depletion. Control or seipin knockdown cells were incubated with oleic acid for indicated times. Targeting of endogenous GPAT4 to LDs during early (**A**) or late (**B**) stage of LD formation were determined by immunofluorescence. Images are presented as projected Z stacks (**A**), or single optical slices (**B**). Red, GPAT4; green, LDs, stained with BODIPY. Bars, 5 μm for (**A**) and 10 μm for (**B**). Note that GPAT4
*Figure 7 continued on next page*

*Figure 7 continued*

targeting to LDs takes place as early as 10 min after adding oleic acid in seipin knockdown. (C) Quantification of GPAT4-positive LDs at 20 min and 8 hr after adding oleic acid. n=30 cells. ***p<0.001. (D) The formation of giant LDs in seipin deficiency depends on LD-localized TG enzymes. Expression of TG synthesis enzymes that are localized to LDs (LD-isoforms) or elsewhere (Non-LD isoforms) were inhibited with specific dsRNAs in combination with control dsRNA or dsRNA against seipin. LD phenotypes were determined 16 hr after LD formation was induced with oleic acid. Bar, 10 µm. Quantification of LD size from 10 cells in each treatment is shown. Lines show mean values. Yellow box indicates giant LDs of diameter > 2.5 µm.

In the current study, we identify a discreet step in LD formation—the conversion of small, nascent LDs (<200 nm diameter) to larger, mature initial LDs (typically 300–500 nm diameter)—and show that seipin acts at this step of LD biogenesis. Oligomers of seipin form highly mobile foci in the ER that interact with small, nascent LDs at ER-LD contact sites and enable their growth to mature iLDs, likely by promoting neutral lipid transport. Without seipin, the precursors of this step, small nascent LDs, accumulate to large numbers adjacent to the ER but fail to grow. With ongoing TG synthesis, some of these intermediates, or possibly other LDs arising from random coalescence of ER TG collections, appear to prematurely engage the eLD pathway, and undergo expansion mediated by LD-localized lipid synthesis enzymes (e.g., GPAT4 and CCT1). As a late phenotype of seipin deficiency, giant LDs (>1–2 µm diameter) form eventually from these fewer abnormal eLDs, probably by coalescence due to a relative lack of PC on their surfaces. This model explains both aspects of the seipin-deficient phenotype—many small LDs and occasional giant LDs—that we and others (*Fei et al., 2011b*, *2008*; *Szymanski et al., 2007*; *Tian et al., 2011*) have observed.

Our data uncover a specific step in LD formation that was previously unrecognized, namely the growth and maturation of nascent LDs to mature iLDs. In wildtype cells, LD formation intermediates at this step are difficult to observe, likely due to their rapid transition through the process. However, with seipin depletion, progression through this step is blocked, resulting in the apparent accumulation of large numbers of intermediates (small, nascent LDs), found in our studies by electron tomography, that fail to mature. These findings indicate that seipin functions downstream of initial lens formation and TG budding, at a more distal step in the maturation of nascent LDs.

How does seipin function in nascent LD maturation? Our studies of the tagged endogenous protein show that seipin localizes to multimeric foci that are highly mobile within the ER, as if scanning the ER for nascent LDs. We also found that some seipin foci encountered and associated with Live-Drop puncta (nascent LDs), became relatively less mobile, and at this point iLD growth ensued. These observations are consistent with the model of seipin engaging nascent LDs and facilitating their growth, likely via transfer of additional neutral lipids, to mature iLDs.

How seipin molecularly interacts with nascent LDs to facilitate LD growth is unknown. Our data are consistent with findings in yeast that implicated a role for seipin oligomers in early stages of LD formation (*Binns et al., 2010*; *Cartwright et al., 2015*). These studies indicated that purified seipin forms oligomers in a toroid-structure comprising approximately 8–12 seipin proteins (*Binns et al., 2010*; *Cartwright and Goodman, 2012*; *Sim et al., 2012*). However, from live-cell imaging, we estimate that the seipin foci include considerably more molecular units in cells (not shown). Possibly seipin oligomers anchored in the ER mediate specific contacts with the surface monolayer of nascent LDs and enable the transfer of lipids (such as TG) to the nascent LDs, allowing them to grow (see model, *Figure 9*). Precisely how seipin mediates this transfer is currently unknown.

Our findings in *Drosophila* and mammalian cells indicate that seipin localizes to ER-LD contact sites and suggest that seipin is part of a protein machinery acting at ER-LD contact sites to promote LD maturation. At later time points in LD biogenesis, we observed that each LD was associated with at least one seipin punctum. Our findings are consistent with several studies in yeast indicating that seipin localizes to ER-LD contact sites (*Cartwright et al., 2015*; *Grippa et al., 2015*; *Wang et al., 2014*). We also found that small nascent LDs almost always associate with the ER, even when seipin was absent, suggesting these LDs are still connected to the ER via contact sites. Consistent with the notion of ER-LD contact sites, the electron-tomograms revealed an absence of ribosomes between the nascent iLDs and the ER membrane. The close connections of the organelles and possible contact sites suggest that other proteins might be involved in establishing ER-LD contact sites and are still present in seipin deficiency.

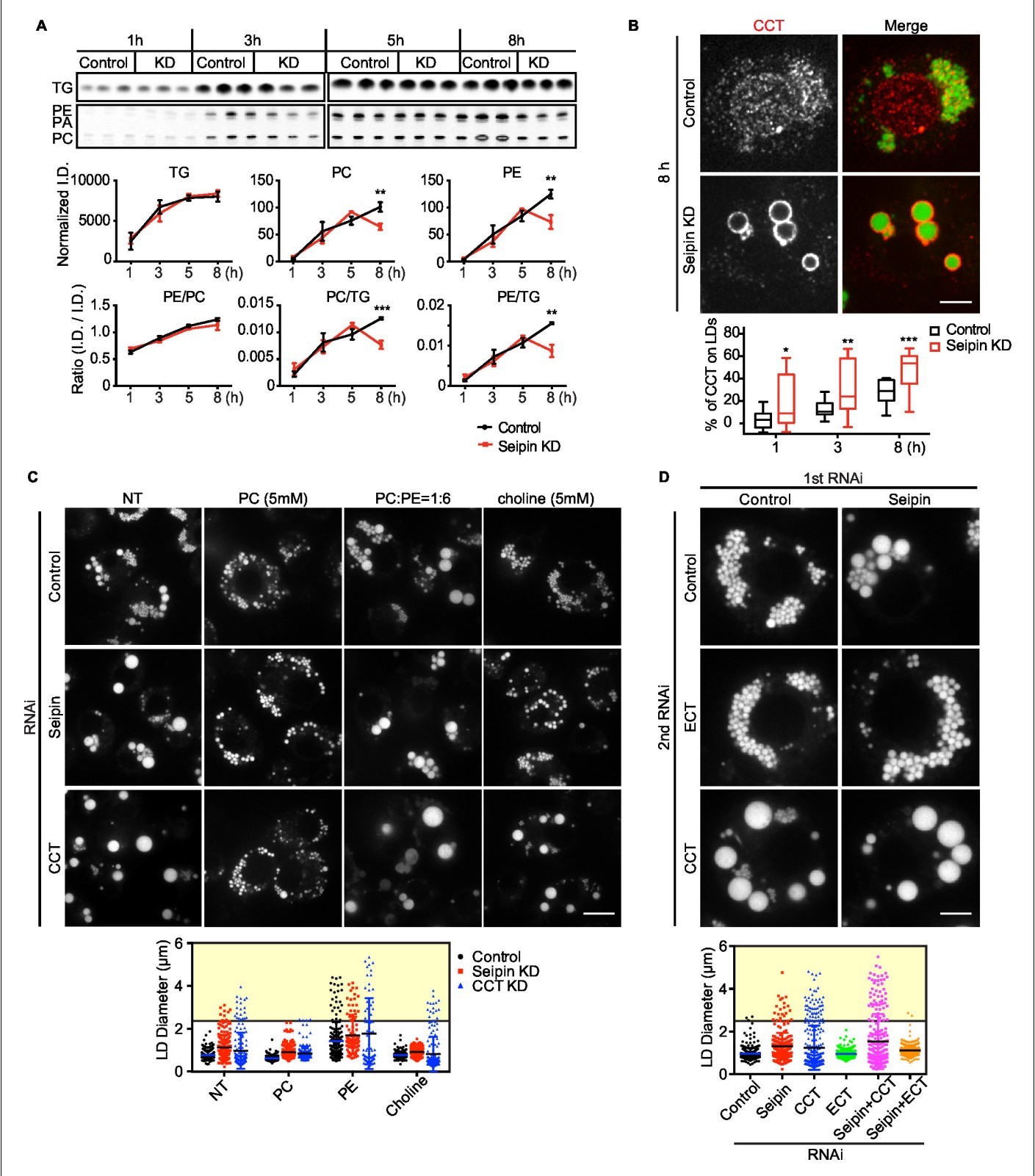

**Figure 8.** Seipin deficiency affects phospholipid composition on LDs at later stages of formation in *Drosophila* S2 cells. (**A**) Phospholipids are deficient on LDs in seipin-knockdown cells 8 hr after initiating formation. Control or seipin knockdown cells were pulse-labeled with [$^{14}$C]-oleic acid (100 μCi/μ mol) for indicated times. LD fractions were purified by gradient centrifugation, and phospholipids and neutral lipids were extracted and separated by TLC. The TLC plate was exposed on an imaging screen, and the intensity of bands was quantified with FIJI. Values are integrated density normalized to

*Figure 8 continued on next page*

*Figure 8 continued*

protein concentration or ratio of integrated density of indicated lipid classes. n=3. **p<0.005; ***p<0.001 (**B**) Targeting of CCT1 to LDs is increased and earlier with seipin depletion. Control or seipin-knockdown cells were incubated with oleic acid for 1, 3, or 8 hr, and localization of endogenous CCT1 was determined by immunofluorescence. Image shows a typical result at 8 hr. Red, CCT1; green, BODIPY. Bar, 5 μm. Targeting of CCT1 to LDs was quantified and is expressed as percentage of CCT1 signal on LDs over total cellular CCT1 signal. Data are presented as box plot with Tukey's test. n=20. *p<0.05; **p<0.01; ***p<0.001. (**C**) Giant LD phenotype in seipin deficiency is rescued by adding PC or choline. Control, seipin, or CCT1 knockdown cells were treated with liposomes containing PC, PE or choline for 24 hr and then incubated with oleic acid for 16 hr. LD phenotype from representative cells are shown. Bar, 10 μm. Quantification of LD size from 10 cells in each treatment is shown. Lines show mean values. Yellow box indicates giant LDs of diameter > 2.5 μm. (**D**) Giant LD phenotype in seipin deficiency is worsened by blocking PC synthesis and ameliorated by blocking PE synthesis. Expression of CCT1 or ECT was inhibited by respective dsRNAs, in addition to control or seipin dsRNAs. Representative LD phenotypes after 16 hr of oleic acid treatment are shown. Bar, 5 μm. Quantification of LD size from 10 cells in each treatment is shown. Lines show mean values. Yellow box indicates giant LDs of diameter > 2.5 μm.

Our findings suggest that seipin's function in converting nascent LDs to mature iLDs is likely its ancient, primary function. In support of this, seipin deficiency resulted in similar phenotypes of larger numbers of LiveDrop-positive foci in cells from flies, human mammary carcinoma cells, and human fibroblasts from seipin-deficient subjects. Additionally, the evolutionarily conserved transmembrane domains and luminal ER loop were sufficient for seipin's function in LD formation. The role of the variable N-terminal domain is less clear. We found that the overexpressed N-terminus of *Drosophila* seipin localized to LDs (*Figure 4A*), suggesting that this part of the protein, though not required for formation, may aid in interaction with LDs. In yeast, the N-terminus was found to be functionally important with respect to the timing of LD formation (*Cartwright et al., 2015*).

Although our findings indicate that seipin functions in an early step in LD formation, they also potentially explain the phenotype of giant LDs found in nearly all cells with seipin deficiency. In the absence of seipin, we found that large numbers of nascent iLDs accumulate, and some of these LDs aberrantly entered the LD expansion pathway. In support of this, lipid synthesis enzymes, such as GPAT4 and CCT1, that are normally found only on late-forming eLDs were aberrantly targeted to iLDs at early time points, and this mislocalization of eLD proteins to iLDs was crucial for the development of the giant LD phenotype at later time points. Similar abnormal protein targeting to LDs was recently found in yeast lacking seipin orthologues (*Grippa et al., 2015*; *Wang et al., 2014*).

Several other models have proposed a primary role for seipin in maintaining ER homeostasis, with associated indirect effects on LD formation. Our data do not support these models. For example, seipin was hypothesized to primarily regulate lipid metabolism in the ER, including phospholipid synthesis (*Fei et al., 2008*, *2011c*; *Tian et al., 2011*), and this in turn affects LD formation. We found no evidence to support this model, either in studies of glycerolipid synthesis rates or in the activities of specific enzymes (such as GPAT; not shown). We also found no evidence of accumulation of PA in membranes of seipin-deficient S2 cells, a finding that has been reported by several groups for seipin deficiency in yeast (*Fei et al., 2011c*; *Sim et al., 2012*; *Tian et al., 2011*; *Wolinski et al., 2015*). We also found no evidence of seipin primarily affecting ER morphology or ER stress activation, which would normally be associated with defects in calcium homeostasis, as has been suggested (*Bi et al., 2014*). Since seipin appears to act downstream of the budding of nascent LDs, it might be expected that ER functions are largely conserved and unaffected by seipin deficiency.

In summary, we provide evidence that seipin functions at a previously unrecognized discrete step in LD biogenesis, enabling nascent LDs to grow to mature iLDs. Cells lacking seipin can still form LDs, but these LDs have irregular size and abnormal lipid and protein composition, which likely leads to cellular dysfunction with respect to storing and reclaiming lipids for cellular needs. In support of this notion, seipin deficiency in humans leads to severe generalized lipodystrophy, with a near absence of adipose mass (*Magré et al., 2001*). Seipin therefore is a crucial part of the cellular protein machinery that serves to organize oil emulsification for fat storage. Our findings also suggest that seipin likely works in concert with other proteins that mediate ER-LD contact to enable the growth of nascent LDs.

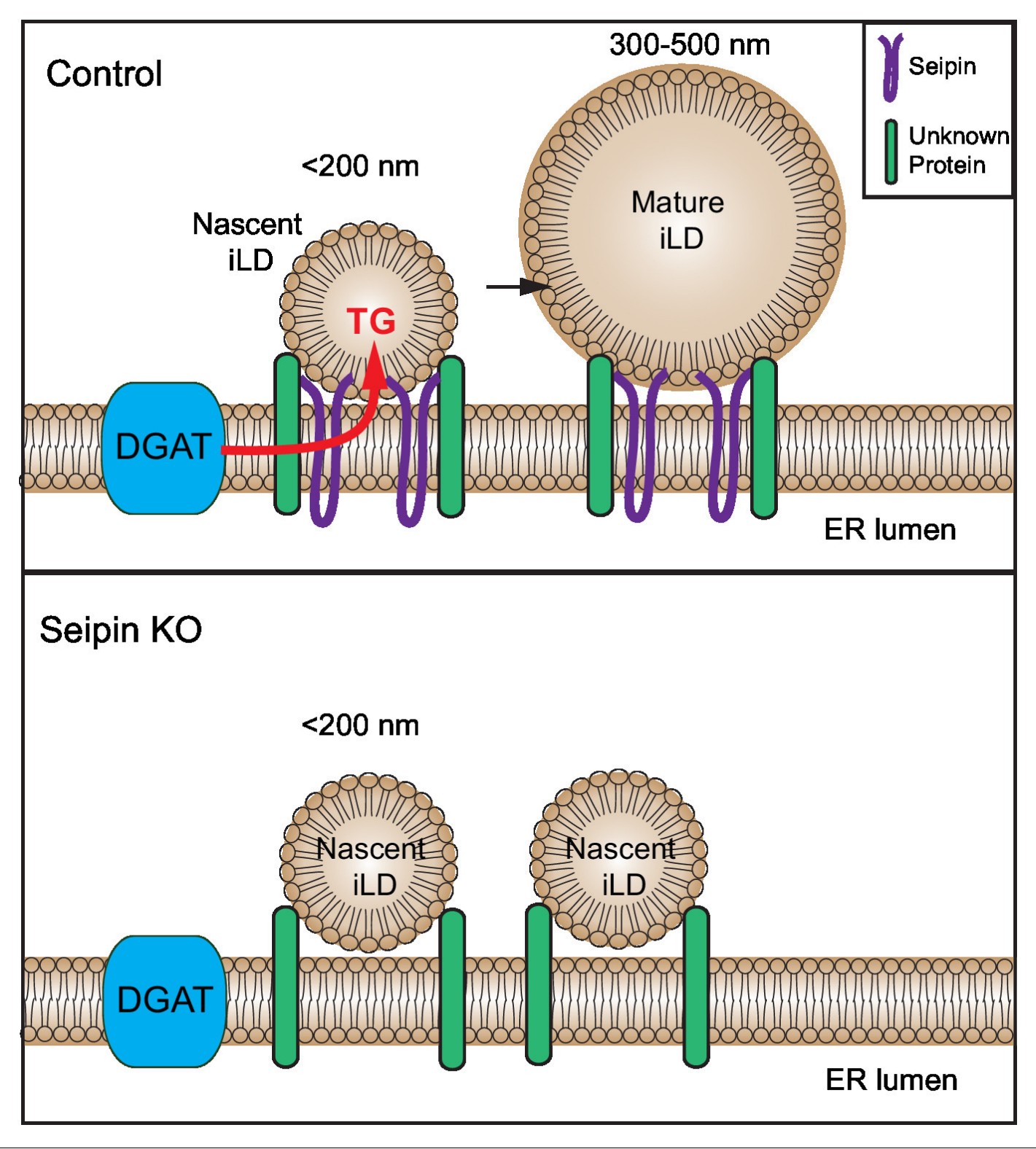

**Figure 9.** Model for the role of seipin in LD formation. Seipin oligomers localize at the contact sites between nascent iLDs and the ER, enabling transfer of lipids (such as TG) to the nascent LDs to convert nascent iLD of <200 nm diameter to mature iLD of 300–500 nm diameter. Without seipin, lipid transfer is inhibited, and LD growth is arrested, leading to the accumulation of nascent iLDs. The maintenance of the contact site between nascent iLDs and the ER does not require seipin; an unknown protein is likely to be involved.

## Materials and methods

### Cell culture, RNAi, and transfection

*Drosophila* S2 cells were kindly provided from the laboratory of Ron Vale (UCSF), SUM159 cells (RRID:CVCL_5423) were from the laboratory of Tomas Kirchausen (Harvard Medical School), and human fibroblasts were from the laboratories of David Savage (Univ. Cambridge) and Abhimanyu Garg (UT Southwestern). *Drosophila* S2 cells were cultured in Schneider's *Drosophila* Medium (Life Technologies, Carlsbad, CA) containing 10% FBS and 50 unit/ml penicillin and 50 µg/ml streptomycin. SUM159 cell were maintained in DMEM/F-12 GlutaMAX (Life Technologies) containing 5% FBS, 100 unit/ml penicillin, and 100 µg/ml streptomycin, 1 µg/ml hydrocortisone (Sigma, St. Louis, MO), 5 µg/ml insulin (Cell Applications, San Diego, CA) and 10 mM HEPES, pH 7.0. Human fibroblasts were cultured in high-glucose DMEM GlutaMAX (Life Technologies) containing 10% FBS, 1 mM sodium pyruvate and 1x NEAA (Life Technologies). Where applicable, cells were incubated with media containing 1 mM oleic acid complexed with 0.5% BSA for various times to induce LD formation.

Knockdown of target genes by RNAi in S2 cells was performed as described (*Krahmer et al., 2011*). Primers used to synthesize dsRNAs for RNAi are listed in *Supplementary file 1*. Cells were analyzed after 4–8 days of RNAi treatment. Transfection of plasmids in S2 cells and mammalian cells was performed with Effectene Transfection Reagent (Qiagen,Valencia, CA) and lipofectamine 3000 (Life Technologies), respectively, according to manufacturer's instructions. Unless otherwise mentioned, transfections were done 18–24 hr before experiment. Plasmids used for transfection are listed in *Supplementary file 1C*.

### Antibodies

Polyclonal antibodies against dGPAT4 (CG3209) and dCCT1 (CG1049) were custom-made (GenScript) and affinity purified (*Krahmer et al., 2011*; *Wilfling et al., 2013*). Mouse polyclonal antibody against hSeipin (A02) was from Abnova (RRID:AB_627320; Taipei City, Taiwan). Antibodies against BiP, Ire1, and CHOP were from Cell Signaling Technology (Danvers, MA). Antibodies against Tubulin (monoclonal) and Calnexin (polyclonal) were from Sigma-Aldrich.

### Generation of GFP-tagged seipin in *Drosophila* cells

A Cas9 and sgRNA co-expression plasmid was generated as described (*Housden et al., 2015*), targeting close to the start codon of the Seipin gene (sgRNA target sequence: GCGCAGCAGGATG TTCATGG). A donor construct was also generated by PCR amplifying homology arms flanking the intended insertion site from genomic DNA extracted from S2R+ cells (genomic coordinates of homology arms (genome release July 2014): X: 2,619,765–2,620,308 and X:2,620,312–2,620,964). These homology arms and GFP coding sequence were assembled into a custom backbone vector (*Housden et al., 2014*) using Golden Gate assembly. Expression and donor constructs were mixed in a 1:1 ratio and transfected into S2R+ cells with Effectene Transfection Reagent (QIAGEN), according to manufacturer's instructions. Four days after transfections, GFP-expressing cells were isolated using fluorescence-activated cell sorting (FACS) and cultured under standard conditions.

### Generating seipin knockouts in SUM159 cells

Cas9 and gRNA expression plasmid (pX459) was obtained from Addgene. The sequence 5′-GAC TAAGGGTGGACGTGATC-3′ was used as a gRNA to direct Cas9 into the exon 3 of the seipin locus. pX459 plasmid (500 ng) was transfected into 80,000 cells, following a published protocol (*Ran et al., 2013*). Briefly, 1 day after transfection, cells were treated with puromycin for 3 days to select for transfected cells. Cells were re-plated into 150 mm dishes at clonal density. Individual colonies were then isolated in 24-well dishes. Screening of positive clones was performed by qPCR (sense primer: 5′- GCATGTTCTTGGTCACCATTTC-3′ and antisense primer 5′-CAAATAGCAGGAGGCTAGA GAAG-3′) using power SYBR green (Life Technologies). Genomic DNA of clones showing mRNA expression defect were extracted (Quick Extract DNA extraction solution; Epicentre), and the genomic sequence surrounding the target exon of seipin were amplified by PCR (sense 5′-GCAAA-GAAGGTGTATGGATGGAC-3′ and antisense 5′-CCGGCCAGTCTCTTAT TACTC-3′). PCR products

were subcloned into a plasmid (Zero Blunt TOPO PCR cloning kit; Life Technologies) to validate the edited region of positive knockout clones by sequencing.

## Immunoblot for seipin

Cells were lysed with lysis buffer (150 mM NaCl, 50 mM Tris-HCl, pH 7.4, 1% Triton X-100) and denatured in Laemmli buffer at 37°C for 10 min. Proteins were separated on 10% SDS-PAGE gel and transferred to a PVDF membrane (BioRad, Hercules, CA). The membrane was blocked with blocking buffer (TBST containing 5% milk) for 24 hr and incubated with *BSCL2* antibody (A02, Abnova) at 1:1000 dilution in blocking buffer for 24 hr. The membrane was then washed with TBST for 10 min x 3 times, and incubated in mouse secondary antibody (Santa Cruz Biotechnology, Dallas, TX) at 1:5000 dilution in blocking buffer. Membrane was washed again with TBST for 10 min x 3 times and revealed using the SuperSignal West Femto kit (ThermoFisher, Waltham, MA).

## RNA extraction and quantitative real-time PCR

Total RNA was isolated using the RNeasy Kit (Qiagen), according to the manufacturer's instructions. Complementary DNA was synthesized using iScript cDNA Synthesis Kit (Bio-Rad), and quantitative real-time PCR (qPCR) was performed in triplicates using SYBR Green PCR Master Mix Kit (Applied Biosystems, Waltham, MA). Sequences of the qPCR primers used are listed in *Supplementary file 1B*.

## Microscopy and image processing

Microscopy was performed on spinning disk confocal (Yokogawa CSU-X1) set up on a Nikon Eclipse Ti inverted microscope with a 100× ApoTIRF 1.4 NA objective (Nikon, Melville, NY) in line with 2x amplification. Fluorophores were excited with 405-, 488-, or 561 nm laser lines, and fluorescence was detected by an iXon Ultra 897 EMCCD camera (Andor, Belfast, UK). Unless otherwise mentioned, bandpass filters (Chroma Technology, ) were applied to all acquisitions. Where applicable, Z stacks of 0.13 µm slices were obtained with piezo Z-stage. For live-cell imaging, temperature, humidity and $CO_2$ were controlled during imaging using a stage-top chamber (Oko Lab).

S2 cells cultured in 24-well plates were transferred to glass-bottom dishes (MatTek Corporation) or glass-bottom plates (In Vitro Scientific) coated with Concanavalin A (Sigma) 1 hr before live-cell imaging or immunofluorescence staining. SUM159 and human fibroblasts were cultured directly in 24-well glass-bottom plates. For live-cell imaging of initial LD formation, equal volumes of media containing 2x oleic acid were added to wells immediately before image acquisition. Where applicable, 0.5 µg/ml BODIPY 493/503 (Life Technologies) was added before and with oleic acid supplementation to stain LDs. To image late LD phenotypes, BODIPY was added 5 min before imaging. For immunofluorescence against dGPAT4 or dCCT1, cells were fixed, permeabilized and immunostained as described (*Wilfling et al., 2013*). Alexa Fluor 555 goat anti-rabbit (Life Technologies) was used as secondary antibody.

Acquired images were processed and quantified manually with FIJI software (http://fiji.sc/Fiji). All quantifications were done on raw images. Where necessary, deconvolution of images was performed using Huygens Professional 15.05 (Scientific Volume Imaging) with CMLE algorithm using a measured PSF for each wavelength.

## Lattice light sheet microscopy

Lattice light sheet microscopy was done as previously described (*Chen et al., 2014*). Control or seipin knockdown *Drosophila* S2 cells expressing Cherry-*LiveDrop* and GFP-sec61 were imaged from 5 mm coverslips using a lattice light-sheet microscope with a square lattice light-sheet. Images were acquired on a Hamamatsu ORCA-Flash 4.0 sCMOS camera, where each plane of the cell was excited sequentially using 488 nm and 560 nm laser lines exposed for 18 ms each, and with an excitation inner/outer numerical aperture of 0.44/0.55 respectively and a corresponding light-sheet length of 10 µm. At each time point, the cells were imagined by scanning the objective and the dithered light-sheet at 200 nm step sizes, thereby capturing a volume of ~50 µm x 50 µm x 20 µm (corresponding to 512 x 512 x 101 pixels) every 4.12 s (which includes a 80 ms pause between time points). The image z-stacks for both channels were obtained between 2–10 min post exposure to 1 mM oleic acid by continuous imaging periods of ~20 min in duration. The deconvolution was performed using

GPU compiled Lucy-Richardson deconvolution algorithm using a measured PSF for each wavelength for 15 iterations.

## Electron microscopy and tomography

Cells in petri dishes were fixed in 2.5% gluteraldehyde in 0.1 M sodium cacodylate buffer, pH 7.4, for 1 hr. Buffer rinsed cells were scraped in 1% gelatin and spun down in 2% agar. Chilled blocks were trimmed and postfixed in 1% osmium tetroxide for 1 hr. The samples were rinsed three times in sodium cacodylate rinse buffer and postfixed in 1% osmium tetroxide for 1 hr. Samples were then rinsed and en bloc stained in aqueous 2% uranyl acetate for 1 hr followed by rinsing, dehydrating in an ethanol series and infiltrated with Embed 812 (Electron Microscopy Sciences) resin and baked over night at 60 C. Hardened blocked were cut using a Leica UltraCut UC7. Sections (60 nm) were collected on formvar/carbon-coated nickel grids and contrast stained with 2% uranyl acetate and lead citrate. They were viewed using an FEI Tencai Biotwin TEM at 80Kv. Images were taken on a Morada CCD using iTEM (Olympus) software.

For tomography, 250 nm sections were collected on formvar/carbon-coated copper grids and contrast stained with 2% uranyl acetate and lead citrate, and 15 nm gold particles were used as fiducial markers. These were viewed using FEI Tecnai TF20 at 200 Kv, rotating angle from 60° to −60°. Data were collected using FEI Eagle 1X1 and reconstructed using IMOD (*Mastronarde, 2008*).

## Adhesive emulsion

Hairpin-cherry-bound lipid droplets were recovered as described (*Kory et al., 2015*). Purified LD solution (20 µl) was added to 4 µl of purified Arf1-alexa488 at 100 µM, and 1 µl of GTP at 10 mM and EDTA (2 mM final concentration), as described (*Thiam et al., 2013a*) to generate the 'aqueous solution'. To prepare the 'triolein solution', 15 µl of DOPC (Avanti) solubilized in chloroform (25 mg/ml) was added to 5 µl of DOPE (Avanti) also in chloroform (25 mg/ml). The chloroform was evaporated under vacuum, and the phospholipid film was re-solubilized in 400 µl of triolein (Avanti) by vortexing. 100 µl of the 'triolein solution' containing phospholipids and 5 µl of the prepared 'aqueous solution' were used to generate water-in-oil droplets (*Kory et al., 2015*) bound by both the hairpin (from purified LDs) and Arf1. Two aqueous drops close enough spontaneously adhere to form a phospholipid bilayer between them. Images of adhering drops were acquired by a Lecia SP5 confocal scanning microscope.

## Subcellular fractionation

Cells were harvested, washed with ice-cold PBS, resuspended in 1 ml of homogenization buffer (250 mM sucrose, 20 mM Tris-HCl, 1 mM EDTA, pH 7.4) in the presence of complete protease inhibitor tablet (Roche), and lysed with a motor-driven Potter-Elvehjem homogenizer (Wheaton). Lysates were sequentially centrifuged at 600 × g for 5 min and 8000 × g for 15 min to remove unbroken cells, nucleus and mitochondria. To isolate microsomes, the 8000 × g supernatant was subject to ultracentrifugation at 100,000 × g for 1 hr. The resulted pellet was then rinsed with TBS (20 mM Tris-HCl, 150 mM NaCl, pH 7.4) and resuspended for analysis. To purify LDs, the 8,000 × g supernatant was mixed with 2 M sucrose (1:1), sequentially overlaid with 1 ml of homogenization buffer and 1 ml of TBS. The gradient was ultracentrifuged at 100,000 × g for 1 hr in a swinging bucket rotor (TLS55, Beckman-Coulter). The fat cake layer (LDs) floating on top the gradient was collected for analysis. Protein concentration was determined by BCA (Thermo Scientific) or Bradford assay (Bio-Rad).

To determine the enrichment of *LiveDrop* in the LD fraction, seipin KO SUM159 cells were transfected with Cherry-*LiveDrop* and incubated with oleate for 1 hr. Cells were then collected and homogenized, and the supernatant from 600 *g*, 5 min centrifugation was adjusted to 30% Optiprep and overlaid with series of Optiprep gradient. After centrifugation, 9 fractions were collected and analyzed by western blotting.

## Metabolic labeling, extraction, and analysis of lipids

To label intracellular lipids with radioactive isotopes, cells were incubated in triplicate with media containing 1 mM [$^{14}$C] oleic acid (25 µCi/µmol) for 0.5, 1, 3, 5, and 8 hr. Labeled cells were lysed by sonication, or fractionated by subcellular fractionation as described above. Lipids from same amount

of protein were extracted with chloroform/methanol (2:1) (*Folch et al., 1957*), separated on silica gel TLC plates (Merck) in a two-solvent system, with CHCl3/methanol/acetic acid/formic acid/$H_2O$ (70:30:12:4:1) for phospholipids and n-heptane/isopropyl ether/acetic acid (60:40:4) for neutral lipids. TLC plates were then exposed to an Imaging Screen-K (Bio-Rad) and revealed with a MPI gel-imaging system (Bio-Rad). The revealed bands were quantified with densitometry.

## Lipid quantification by lipidomics

To quantify lipids in control and seipin knockout SUM159 cells with LC-MS/MS, lipids were extracted with chloroform/methanol (2:1) as described (*Folch et al., 1957*). Extracted lipids were first separated on an Accucore C18 column (2.1 x 150 mm, 2.6 μm; Thermo) connected to an Ultimate 3000 HPLC (Thermo), using a binary solvent system (mobile phase A: 50:50 ACN/$H_2O$, 10 mM $NH_4HCO_2$, 0.1% formic acid; mobile phase B: 10:88:2 ACN/IPA /$H_2O$, 2 mM $NH_4HCO_2$, 0.02% formic acid). The HPLC was connected on-line to a Q-exactive mass spectrometer equipped with an electrospray ionization source (ESI; Thermo). Mass spectra were acquired in data-dependent mode to automatically switch between full scan MS and up to 15 data-dependent MS/MS scans. The 15 most intense ions from the survey scan were selected and fragmented with higher energy collision dissociation (HCD) with stepped normalized collision energies of 20, 30 and 40 in negative mode and 25 and 35 in positive ion mode. Peaks were analyzed using the Lipid Search algorithm (MKI, Tokyo, Japan), defined through raw files, product ion and precursor ion accurate masses. Candidate molecular species were identified by database (>1,000,000 entries) search of positive and negative ion adducts. Mass tolerance was set to 5 ppm for the precursor mass. Samples were aligned within a time window and results combined in a single report. Internal standards spiked in prior to extraction were used for normalization and calculation of the molar amounts of lipids.

To measure metabolic flux in S2 cells with shotgun mass spectrometry, cells were pulse labeled with [$^{13}C_5$]-oleic acid (Sigma) for 3 hr. Cells were then homogenized and lipids were extracted. Quantification of lipids by high-resolution shotgun mass spectrometry was performed as described (PMID: 25391725) (*Ejsing et al., 2009*).

## Statistical analyses

Data are presented as means ± SD unless otherwise stated. Statistical significance was evaluated by two-tailed Student's t-test or paired t-test and plotted with GraphPad Prism 6 software.

## Acknowledgements

We thank Xun Huang (Chinese Academy of Sciences) for hSeipin cDNA, Stephanie Mohr of the *Drosophila* RNAi Screening Center at HMS (supported by NIH NIGMS R01 GM067761) and Gerry Marsischky of the HMS Genome Engineering Production Group (supported by the HMS Tools and Technology program) for reagents and advice, Zon Weng Lai, Manuele Piccolis, Zhihuan Li, and Nora Kory (Harvard T H Chan School of Public Health) for their help with experiments and discussions, Guangwei Du (University of Texas Health Science Center) for providing the GFP-PASS plasmid, Jan Hoffmann (Carnegie Mellon University) for help with ER morphology evaluation, and Gary Howard for editorial assistance.

## Additional information

### Funding

| Funder | Grant reference number | Author |
| --- | --- | --- |
| Canadian Institutes of Health Research | Fellowship Award | Huajin Wang |
| European Molecular Biology Organization | Longterm Fellowship EMBOLFT355 | Michel Becuwe |
| Wellcome Trust | WT107064 | David B Savage |
| NIHR Cambridge Biomedical Research Centre | | David B Savage |

| | | |
|---|---|---|
| Howard Hughes Medical Institute | | Norbert Perrimon<br>Tobias C Walther |
| Biogen | | Tomas Kirchhausen |
| National Institutes of Health | GM-075252 | Tomas Kirchhausen |
| Villum Fonden | VKR023439 | Christer S Ejsing |
| Strategiske Forskningsråd | 11-116196 | Christer S Ejsing |
| National Institutes of Health | GM097194 | Tobias C Walther |
| G Harold and Leila Y. Mathers Foundation | | Tobias C Walther |
| National Institutes of Health | GM099844 | Robert V Farese Jr |

The funders had no role in study design, data collection and interpretation, or the decision to submit the work for publication.

### Author contributions

HW, Conception and design, Acquisition of data, Analysis and interpretation of data, Drafting or revising the article; MB, BEH, MMG, ART, FF, SU, Acquisition of data, Analysis and interpretation of data; CC, Acquisition of data, Contributed unpublished essential data or reagents; AJP, M-JO, QL, HKH-B, Acquisition of data; XNL, TK, Analysis and interpretation of data; DBS, AKA, AG, NP, Contributed unpublished essential data or reagents; CSE, Conception and design, Analysis and interpretation of data; TCW, RVF, Conception and design, Analysis and interpretation of data, Drafting or revising the article

### Author ORCIDs

Robert V Farese Jr, http://orcid.org/0000-0001-8103-2239

## Additional files

### Supplementary files

• Supplementary file 1. List of primers and plasmids. (A) Sequences of primers used to generate dsRNA. (B) Sequences of primers used for qPCR. (C) List of plasmids used.

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
