## [Decision Letter]

Thank you for submitting your article "Seipin is required for converting nascent to mature lipid droplets" for consideration by *eLife*. Your article has been favorably evaluated by Randy Schekman (Senior editor) and four reviewers, one of whom is a member of our Board of Reviewing Editors. The following individuals involved in review of your submission have agreed to reveal their identity: Rudolph Zechner and Joel Goodman.

The reviewers have discussed the reviews with one another and the Reviewing Editor has drafted this decision to help you prepare a revised submission. We hope you will be able to submit the revised version within two months, so please let us know if you have any questions first. Given the scope of the criticisms, it may be best if you were able to respond with an action plan and time table for submission of a revised manuscript.

Summary:

The laboratory of Robert Farese and Tobi Walther have been interested in the biogenesis of lipid droplets in insect cells and mammalian cells. They have been leaders in understanding the roles of triglyceride biosynthetic enzymes. In the current manuscript, they have worked to define the cell biology of seipin, an enigmatic protein that is mutated in a subset of patients with generalized lipodystrophy. In this paper, they provide evidence that seipin is involved in the maturation of small nascent lipid droplets to mature lipid droplets. This concept had already been proposed in yeast, but the manuscript by Farese and Walther elucidates the concept more clearly. They report that seipin forms discrete and dynamic foci in the ER that this molecule interacts with nascent LDs, and that this interaction is important for the growth and maturation of lipid droplets. In the absence of seipin, they propose that immature lipid droplets accumulate, and that the occasional lipid droplet that happens to grow is abnormal, in that it prematurely acquires lipid biosynthetic enzymes.

This manuscript was reviewed by four reviewers. All recognized that the topic was important; all were positive to varying degrees about the manuscript and thought that it advanced our understanding of seipin cell biology. However, two of the reviewers had reservations. One of the reviewers was disappointed by the electron microscopy studies. In considering the original manuscript, it was the hope of the reviewers that the authors would characterize LiveDrops at the EM level. Such an undertaking would presumably require APEX electron microscopy. In the revised manuscript, the authors noted the size of lipid droplets in seipin-deficient cells, but the relationship of those lipid droplets to LiveDrops or to seipin is not clear. The authors noted that there appeared to be an accumulation of 200-nm lipid droplets in the seipin-deficient cells, but the data were not convincing because it was not clear how many experiments were performed with how many cell lines-and under what conditions. Also, it was not clear whether a similar accumulation of 200-nm particles occurs in tissues of seipin-deficient mice. Also, one reviewer was not convinced by the "filaments" in the EM tomography studies. Were those filaments only present in the interface between the lipid droplet and the ER, or did they surround the lipid droplet? Finally, several reviewers were disappointed by quantification of findings in several of the figures.

Essential revisions:

1) The authors should use electron microscopy to characterize LiveDrops, and not simply characterize the size of lipid droplets in seipin-deficient cells. Alternatively, the authors should make a far more convincing case for the accumulation of 200-nm particles in seipin-deficient cell lines. The latter approach would need to include data on whether an accumulation of 200-nm particles occurs in specific tissues of seipin-deficient mice, for example in skeletal muscle or heart.

2) The authors should include further EM tomography studies and address whether the putative "filamentous structures" are truly unique to the interface between the ER and the lipid droplet.

3) Explain why the GPAT4-based LiveDrop construct preferentially binds to nascent lipid droplets but that GPAT4 apparently has a preference for large lipid droplets.

4) As the authors acknowledge, several independent research groups have demonstrated PA accumulation in yeast seipin (Fld1) mutants, which was not seen in the cells studied here. However, the authors do not present a strong explanation. Is this attributable to a species difference between yeast and *Drosophila* cells, or is there another difference in the measurements performed in the current study? It would be enlightening for the authors to compare seipin-deficient yeast cells to settle this once and for all.

5) The model proposed in Figure 9 is not satisfying, as it does not convey new information that was uncovered in this study, and includes a component ("unknown protein") that is not discussed. The figure should better represent findings presented here, such as seipin mobility and accumulation of glycerolipid biosynthetic proteins on some seipin-deficient LDs.

6) Some statements are not backed by rigorous data. For example, statistical analyses should be included in Figure 7, Figure 8.

7) The authors use multiple cell types, but it is not clear by reading some Figure legends which cell types are used. It would be helpful to add more transparency by referring to either "S2 cells" or mammalian cells at the beginning of Figure 2, Figure 4, Figure 6, Figure 7 and Figure 8.

[Editors’ note: a previous version of this study was rejected after peer review, but the authors submitted for reconsideration. The previous decision letter after peer review is shown below.]

Thank you for choosing to send your work, "Seipin organizes neutral lipids in the ER for initial lipid droplet formation", for consideration at *eLife*. Your initial submission has been assessed by Vivek Malhotra in consultation with a member of the Board of Reviewing Editors (BRE). The BRE member sent your manuscript to three internationally recognized experts in lipid metabolism for detailed review. The three external reviewers then discussed their reviews and their recommendations with Dr. Vivek and the BRE member.

Although the work deals with an extremely important topic in the cell biology of lipid droplet biogenesis, we regret to inform you that we will not be able to publish your work in *eLife*. All three reviewers agreed that the findings were too preliminary. Each of the reviewers had substantive concerns, and it seemed unlikely that these concerns could be resolved in a reasonable period of time. All three external reviewers were duly impressed by the morphological findings detected with the LiveDrop probe, but there was a consensus that the nature of the "LiveDrop" needs to be characterized at the ultrastructural level. The reviewers also thought that it would be important to demonstrate that the LiveDrop probe detects equivalent structures in wild-type cells and seipin-deficient cells. One of the reviewers was not fond of the "LiveDrop" nomenclature, particularly since the primordial lipid droplets have not been fully characterized at an ultrastructural level. Another concern is that the manuscript, while strong on immunofluorescence microscopy, did not reveal mechanisms of seipin activity. Finally, one of the reviewers was not satisfied with the biochemistry and thought that the conclusion that there was no direct effect of seipin on phospholipid synthesis might be premature, stating "the data shows that PC levels in LDs and the incorporation of FA into LD-associated PC are decreased in seipin deficient cells, despite abundant presence of CCT on LDs. This argues for a specific defect in LD-PC synthesis due to enzyme inhibition (e.g. CPT) or choline deficiency. A more elaborate lipid analysis of LD-associated lipids and ER lipids and better characterization of the involved biochemical pathways are needed to conclude that seipin does not interfere with phospholipid metabolism. Specifically, the role of seipin in PA metabolism has not been sufficiently addressed to conclude that it plays no role in PA metabolism (as claimed by Fei et al., Tian et al. or Wolinsky et al.). Cellular concentrations of PA in cell homogenates, microsomal fractions, or LDs are extremely low. Regular lipidomic approaches may not be adequate to detect differences (e.g. Figure 7—figure supplement 1D). The authors should consider using PA specific probes such as the GFP-Spo20 reporter protein." The reviewers’ comments are pasted at the bottom of the letter.

Please note that we aim to publish articles with a single round of revision that would typically be accomplished within two months. While your observations have great potential, it seems unlikely that the necessary revisions could be completed within a few months.

We do not intend any criticism of the quality of your data. We wish you the very best of luck with your work, and we sincerely hope that you will consider *eLife* for future submissions.

Reviewers comments (verbatim)

*Reviewer 1:*

Wang et al. present an interesting study to show that seipin acts early during lipid droplet (LD) formation as a "triglyceride trapping" agent. The authors further demonstrate that this process is important for the appropriate assembly of lipid synthesis enzymes and controlled maturation of LDs. Although the actual biochemical function of seipin remains to stay undefined, the study adds important new information to our understanding of LD formation. The experimental approach appears very robust and adequate to address challenging biology.

Only a few open points need to be addressed:

1) Wouldn't it be more appropriate to name "initial LDs" "nascent LDs"?

2) The interpretation that seipin may not interfere with phospholipid synthesis appears premature. In fact, the current study shows that PC synthesis is decreased despite abundant presence of CCT on LDs. Doesn't this argue for a potential interference of seipin deficiency with the subsequent CPT reaction? Or does seipin deficiency lead to an undersupply of choline?

3) Due to the extremely low concentrations of PA in cell homogenates and microsomal fractions, regular lipidomic approaches may not be sufficient to detect differences (e.g. Figure 7). The authors should consider using PA specific probes such as GFP-Spo20 reporter protein.

4) How do the authors explain the conflicting data on the UPR in response to seipin deficiency compares to observations in previous studies?

Reviewer 2:

Wang and colleagues use *Drosophila* cultured cells to study the function of Seipin, a conserved protein required for normal lipid droplets and that has disease relevance. Despite the vast literature on seipin its molecular function is not known yet. Here the authors find that depletion of seipin in *Drosophila* induces LD morphology defects. It is shown that abnormal droplets in seipin-deficient cells are sensitive to changes in phospholipid synthesis without affecting lipid levels, largely confirming previous work. Then, the authors use "LiveDrop", presumably a very sensitive marker for LDs to interrogate early stages of LD formation. In wt cells, LiveDrop preceded BODIPY staining in nascent LDs. In contrast, dynamic LiveDrop puncta that remain BODIPY-negative were detected in seipin-depleted cells. The appearance of these puncta was diminished upon interference with TAG synthesis. These results led to the proposal that seipin organizes initial TAG lenses into nascent LDs. These are potentially the most exciting findings but in my opinion, they are strongly over interpreted and undocumented. It is speculated that LiveDrop puncta mark initial TAG lenses but it not at all clear what they are. A thorough characterization of the LiveDrop puncta is essential to support this model- this would require both biochemical and ultrastructural analysis. If small puncta in seipin deficient cells are incipient lenses too small to be detected by BODIPY, it is expected that bigger puncta should become BODIPY-positive. However this is not the case for example in Figure 2 – at 22min LiveDrop puncta has a diameter of almost 0.5 μm (comparable to control) but it is still BODIPY-negative.

Reviewer 3:

Wang et al. report studies, mainly in *Drosophila* S2 cells on the function of seipin. The authors first show the generation of "supersized" droplets in seipin knock-down cells upon adding oleate to the medium. This is not due to any significant change in total lipid composition or amounts. Next, they generate a probe (LiveDrop) and show that it can detect very early neutral lipid (NL) aggregates. In wild type cells these are converted to droplets (BODIPY stained), but in the seipin KD the aggregates convert much less often. They show that the aggregates are highly mobile in the KD. Using CRISPR, they next show that seipin is itself mobile until it binds to an aggregate; its movement slows after that. The authors next show that GPAT4 targets to droplets much more rapidly in the KD, which can account for the supersized phenotype. They show that these supersized droplets have less phospholipid, as expected from geometry. They then do a seipin deletion analysis on CRISPR KO cells and show that the loop and TM regions of seipin are sufficient for function. Last, they show similar results in mammalian cells, including BSCL2- cells from a patient.

The overall finding is that seipin catalyzes the conversion of tiny nascent neutral lipid aggregates in the ER into nascent droplets. Our group in yeast has shown a role of seipin in droplet initiation and morphology. This work here confirms and vastly extends these results, using animal cells. They key new findings are: (1) development of a novel probe (LiveDrop) that detects tiny neutral lipid aggregates before they are visible with BODIPY as bona fide droplets; (2) a convincing demonstration that seipin blocks conversion of these aggregates to droplets. Thus, the authors have narrowed down the step in which seipin works; (3) ruling out effects on phospholipid metabolism as being important (contrary to what many in the field thought; (4) evidence that "supersized" droplets often seen in seipin knockout strains is due to strong targeting of GPAT 4 to droplets as well as a transient lack of phospholipids; and (5) the role of seipin in converting the lipid aggregates to droplets is shared in mammalian cells. Most of the work is based on fluorescence microscopy, which is of high quality and appropriately quantified.

So, while this work does not propose a novel mechanism for seipin function, it strongly reinforces the idea that seipin works on an early phase of droplet formation as a conserved mechanism, provides strong evidence for the lens model of droplet formation (previously only a theory), and identifies a discrete step for seipin action: conversion of the lens (seen with the use of LiveDrop) to a bona fide droplet. The work is of cutting edge quality, and there is enough here to be of interest to the general readership of *eLife*.

---

## [Author Response]

The laboratory of Robert Farese and Tobi Walther have been interested in the biogenesis of lipid droplets in insect cells and mammalian cells. They have been leaders in understanding the roles of triglyceride biosynthetic enzymes. In the current manuscript, they have worked to define the cell biology of seipin, an enigmatic protein that is mutated in a subset of patients with generalized lipodystrophy. In this paper, they provide evidence that seipin is involved in the maturation of small nascent lipid droplets to mature lipid droplets. This concept had already been proposed in yeast, but the manuscript by Farese and Walther elucidates the concept more clearly. They report that seipin forms discrete and dynamic foci in the ER that this molecule interacts with nascent LDs, and that this interaction is important for the growth and maturation of lipid droplets. In the absence of seipin, they propose that immature lipid droplets accumulate, and that the occasional lipid droplet that happens to grow is abnormal, in that it prematurely acquires lipid biosynthetic enzymes.

We thank the editor and reviewers for their time and the critical and helpful evaluation of our manuscript. This is an excellent summary of the paper. However, we think that it is inaccurate to say that the concept of seipin helping nascent LDs in conversion to mature LDs has been proposed in yeast. The published yeast studies established that LD formation is delayed in seipin deficiency and that LD formation is affected, but did not provide data showing that seipin facilitates a specific step in LD formation. We believe our findings go considerably beyond the published yeast studies in this respect and provide new insights into the function of seipin in LD formation. Also, yeast and mammalian LDs and TG synthesis are considerably different. A key enzyme for LD formation in mammals, DGAT1, is absent in yeast and it is very much unclear to date whether the processes and seipin function are identical between systems. Thus, we feel it is justified and important to independently investigate seipin function in different model systems, in addition to yeast.

This manuscript was reviewed by four reviewers. All recognized that the topic was important; all were positive to varying degrees about the manuscript and thought that it advanced our understanding of seipin cell biology. However, two of the reviewers had reservations. One of the reviewers was disappointed by the electron microscopy studies. In considering the original manuscript, it was the hope of the reviewers that the authors would characterize LiveDrops at the EM level. Such an undertaking would presumably require APEX electron microscopy. In the revised manuscript, the authors noted the size of lipid droplets in seipin-deficient cells, but the relationship of those lipid droplets to LiveDrops or to seipin is not clear. The authors noted that there appeared to be an accumulation of 200-nm lipid droplets in the seipin-deficient cells, but the data were not convincing because it was not clear how many experiments were performed with how many cell lines-and under what conditions. Also, it was not clear whether a similar accumulation of 200-nm particles occurs in tissues of seipin-deficient mice. Also, one reviewer was not convinced by the "filaments" in the EM tomography studies. Were those filaments only present in the interface between the lipid droplet and the ER, or did they surround the lipid droplet? Finally, several reviewers were disappointed by quantification of findings in several of the figures.

We address each of these points in the comments to specific points below.

Essential revisions:

1) The authors should use electron microscopy to characterize LiveDrops, and not simply characterize the size of lipid droplets in seipin-deficient cells. Alternatively, the authors should make a far more convincing case for the accumulation of 200-nm particles in seipin-deficient cell lines. The latter approach would need to include data on whether an accumulation of 200-nm particles occurs in specific tissues of seipin-deficient mice, for example in skeletal muscle or heart.

We understand the sentiment behind the comments, but we respectfully disagree about the utility of the suggested experiments. We understand that the question behind this point is whether LiveDrop is bona fide marker for Lipid Droplets. This is clearly the case because:

1) LiveDrop represents the membrane anchoring sequence of an enzyme (GPAT4) targeted to the LD surface (Wilfling et al., Dev.Cell 2013);

2) In a time course, all LiveDrop puncta acquire hallmarks of LDs, i.e. BODIPY staining, in wild-type cells;

3) LiveDrop localization to puncta is dependent on TG synthesis;

4) LiveDrop partitions to monolayer over bilayer surfaces in vitro;

5) Accumulation of LiveDrop puncta in seipin-deficient cells corresponds with the accumulation of small, nascent LDs in electron microscopy.

We could take some months to show by EM that LiveDrop localizes in part to nascent LDs, but this would not, in our minds, add to the paper. As LiveDrop co-localizes to both the ER and nascent LDs, we do not think such experiments would adequately answer the question (e.g., we would need to show accumulation of LiveDrop in LDs over ER, which is very hard to do at a quantitative level by EM).

Instead, our study shows that lipid accumulation of TG, as detected by the LiveDrop probe, are found with seipin deficiency in *Drosophila* cells, human mammary carcinoma cells, and patient cells with seipin deficiency. We elected to perform thin-section EM and electron tomography on the seipin knockout mammary carcinoma cells. We found that the TG accumulations correspond to highly abundant, nascent (<200 nm diameter) LDs. We counted these in sections from at least five cells and found a mean diameter of 160 nm with very small standard deviation (~20 nm). Thus, we believe that the massive accumulation of small LDs observed by EM (we will add a gallery of cells to show the dramatic effect in overview) is the only plausible explanation for the similar, massive accumulation of LiveDrop foci observed by light microscopy.

What we can offer to do in response is to show that isolation of these nascent LDs from seipin knockout cells reveals that LiveDrop protein is localized to fractions with the nascent LDs. This will take us a week or two to complete.

We can also further quantify the numbers of nascent LDs in seipin deficient cells. We already performed EM on these cells and counted many LDs. These data are shown in Figure 5. Although we could count more LDs, we are doubtful that more counting will alter our conclusions. Nevertheless, we can do this.

We also disagree with the utility of examining LDs in tissues of seipin-deficient mice. LD phenotypes due to *seipin* deficiency has been reported in a large number of model systems and studies. However, because in these systems there is no synchronized LD formation, the early and likely most direct role of seipin has been a mystery. To overcome this, we are studying the earliest stages of LD formation in cell models because cells are the best living model system to examine this problem in time course experiments. Even in this system, the late effects (i.e., > 4-6 hours) are, as mentioned, the formation of large LDs due to expansion and localized TG synthesis enzymes. Any studies of tissues of the knockout mice would be hugely complicated by the late phenotype of giant LD formation. We cannot see a useful experiment to look at the earliest stages of LD formation in the knockout mouse tissues. Not to mention, such a study would require a huge experimental effort (likely in the order of a year of work) as *seipin* deficient mice are currently not available to us.

2) The authors should include further EM tomography studies and address whether the putative "filamentous structures" are truly unique to the interface between the ER and the lipid droplet.

We thank the reviewers for the suggestion, but we respectfully disagree. The current electron tomography studies are suggestive of a filamentous structure. We can add another example of such a LD-ER area. Further electron tomography will not clarify this, however (we can add more examples but this is limited due to the time consuming data acquisition). It will instead be necessary to identify the protein, test for its requirement, etc., which is clearly a separate project beyond the scope of the current publication.

However, since the interpretation of the current data is speculative with respect to filamentous structure, we prefer instead to remove this speculative interpretation and simply mention in the Results that the electron tomography revealed areas of density between the organelles, suggestive of a contact zone.

3) Explain why the GPAT4-based LiveDrop construct preferentially binds to nascent lipid droplets but that GPAT4 apparently has a preference for large lipid droplets.

This is an excellent question and one that we are currently investigating. Our preliminary data indicate that indeed LiveDrop localizes to all LDs at the time of formation whereas full-length GPAT4 localizes only to the eLD population. In contrast, in wildtype cells full length GPAT4 is excluded from nascent LDs and requires the Arf1/COPI machinery for targeting to eLDs at a later time point (Wilfling et al., *eLife* 2014). This suggests the hypothesis that full length GPAT4 must be retained in the ER during LD formation due to properties of the rest of the protein (i.e., not the membrane hairpin domain). We are testing this hypothesis. For clarity, we will explain more explicitly the different behavior of the full-length vs. hairpin in the revised version.

*4) As the authors acknowledge, several independent research groups have demonstrated PA accumulation in yeast seipin (Fld1) mutants, which was not seen in the cells studied here. However, the authors do not present a strong explanation. Is this attributable to a species difference between yeast and Drosophila cells, or is there another difference in the measurements performed in the current study? It would be enlightening for the authors to compare seipin-deficient yeast cells to settle this once and for all.*

We acknowledge that there are apparent differences between what is found in yeast and *Drosophila* cells regarding PA and seipin knockout. This likely results from species differences in lipid metabolism. Our data, however, show that the seipin knockout phenotype with respect to LD formation occurs without changes in PA in both *Drosophila* and mammalian cell models, effectively excluding this as the cause for the formation abnormality we found in our system. We think that PA could accumulate as a consequence of seipin deficiency in yeast. We respectfully disagree on the utility of our measuring PA levels in yeast—whether we find elevations or not will not settle the issue.

5) The model proposed in Figure 9 is not satisfying, as it does not convey new information that was uncovered in this study, and includes a component ("unknown protein") that is not discussed. The figure should better represent findings presented here, such as seipin mobility and accumulation of glycerolipid biosynthetic proteins on some seipin-deficient LDs.

We will remove the model from the figure set.

6) Some statements are not backed by rigorous data. For example, statistical analyses should be included in Figure 7, Figure 8.

We will add statistical analyses for these data.

7) The authors use multiple cell types, but it is not clear by reading some Figure legends which cell types are used. It would be helpful to add more transparency by referring to either "S2 cells" or mammalian cells at the beginning of Figure 2, Figure 4, Figure 6, Figure 7 and Figure 8.

This will be clarified in the revised version.

[Editors’ note: the author responses to the previous round of peer review follow.]

Although the work deals with an extremely important topic in the cell biology of lipid droplet biogenesis, we regret to inform you that we will not be able to publish your work in eLife. All three reviewers agreed that the findings were too preliminary. Each of the reviewers had substantive concerns, and it seemed unlikely that these concerns could be resolved in a reasonable period of time. All three external reviewers were duly impressed by the morphological findings detected with the LiveDrop probe, but there was a consensus that the nature of the "LiveDrop" needs to be characterized at the ultrastructural level. The reviewers also thought that it would be important to demonstrate that the LiveDrop probe detects equivalent structures in wild-type cells and seipin-deficient cells. One of the reviewers was not fond of the "LiveDrop" nomenclature, particularly since the primordial lipid droplets have not been fully characterized at an ultrastructural level. Another concern is that the manuscript, while strong on immunofluorescence microscopy, did not reveal mechanisms of seipin activity. Finally, one of the reviewers was not satisfied with the biochemistry and thought that the conclusion that there was no direct effect of seipin on phospholipid synthesis might be premature, stating "the data shows that PC levels in LDs and the incorporation of FA into LD-associated PC are decreased in seipin deficient cells, despite abundant presence of CCT on LDs. This argues for a specific defect in LD-PC synthesis due to enzyme inhibition (e.g. CPT) or choline deficiency. A more elaborate lipid analysis of LD-associated lipids and ER lipids and better characterization of the involved biochemical pathways are needed to conclude that seipin does not interfere with phospholipid metabolism. Specifically, the role of seipin in PA metabolism has not been sufficiently addressed to conclude that it plays no role in PA metabolism (as claimed by Fei et al., Tian et al. or Wolinsky et al.). Cellular concentrations of PA in cell homogenates, microsomal fractions, or LDs are extremely low. Regular lipidomic approaches may not be adequate to detect differences (e.g. Figure 7—figure supplement 1D). The authors should consider using PA specific probes such as the GFP-Spo20 reporter protein." The reviewers’ comments are pasted at the bottom of the letter.

Please note that we aim to publish articles with a single round of revision that would typically be accomplished within two months. While your observations have great potential, it seems unlikely that the necessary revisions could be completed within a few months.

We do not intend any criticism of the quality of your data. We wish you the very best of luck with your work, and we sincerely hope that you will consider eLife for future submissions.

Reviewers comments (verbatim)

Reviewer 1:

Only a few open points need to be addressed:

1) Wouldn't it be more appropriate to name "initial LDs" "nascent LDs"?

2) The interpretation that seipin may not interfere with phospholipid synthesis appears premature. In fact, the current study shows that PC synthesis is decreased despite abundant presence of CCT on LDs. Doesn't this argue for a potential interference of seipin deficiency with the subsequent CPT reaction? Or does seipin deficiency lead to an undersupply of choline?

3) Due to the extremely low concentrations of PA in cell homogenates and microsomal fractions, regular lipidomic approaches may not be sufficient to detect differences (e.g. Figure 7). The authors should consider using PA specific probes such as GFP-Spo20 reporter protein.

4) How do the authors explain the conflicting data on the UPR in response to seipin deficiency compares to observations in previous studies?

Reviewer 2:

Wang and colleagues use Drosophila cultured cells to study the function of Seipin, a conserved protein required for normal lipid droplets and that has disease relevance. Despite the vast literature on seipin its molecular function is not known yet. Here the authors find that depletion of seipin in Drosophila induces LD morphology defects. It is shown that abnormal droplets in seipin-deficient cells are sensitive to changes in phospholipid synthesis without affecting lipid levels, largely confirming previous work. Then, the authors use "LiveDrop", presumably a very sensitive marker for LDs to interrogate early stages of LD formation. In wt cells, LiveDrop preceded BODIPY staining in nascent LDs. In contrast, dynamic LiveDrop puncta that remain BODIPY-negative were detected in seipin-depleted cells. The appearance of these puncta was diminished upon interference with TAG synthesis. These results led to the proposal that seipin organizes initial TAG lenses into nascent LDs. These are potentially the most exciting findings but in my opinion, they are strongly over interpreted and undocumented. It is speculated that LiveDrop puncta mark initial TAG lenses but it not at all clear what they are. A thorough characterization of the LiveDrop puncta is essential to support this model- this would require both biochemical and ultrastructural analysis. If small puncta in seipin deficient cells are incipient lenses too small to be detected by BODIPY, it is expected that bigger puncta should become BODIPY-positive. However this is not the case for example in Figure 2 – at 22min Livedrop puncta has a diameter of almost 0.5 μm (comparable to control) but it is still BODIPY-negative. Besides this major concern on the characterization of the "LiveDrop puncta" there are a number of other minor issues that it would be important to address to make this work suitable for publication.

Based on the reviewer’s excellent comments, we have performed additional experiments, as well as more rigorous characterization of some reagents we use. These experiments lead us to a breakthrough in our thinking concerning seipin function. In particular added electron tomography of seipin knockout cells revealed that the accumulating intermediates are nascent LDs that remain in contact, but are separate, from the ER. With these data, we now hypothesize that seipin functions at ER-LD contact sites to enable lipid transfer to the nascent LDs, enabling their growth to mature initial LDs. This is a major advance in understanding seipin and LD formation and is reflected in many changes in the new manuscript, including the new title. In addition, we have added significant amounts of data for the characterization of LiveDrop, the probe we use to visualize nascent LDs, and we have added additional data regarding our lipid measurements. For the issue of phospholipids changes in seipin-depleted cells, we reviewed our lipidomic data and concluded that our assays are linear for detection of PA in the experiments. In addition, we performed the experiment of expressing a “PA probe” in wild-type and seipin KO SUM cells, and we did not see accumulations of this probe near the nascent LDs (as others have seen in yeast with Fld1 deletion). We agree with the reviewer that at late time points, there are LD-specific changes in phospholipids, including our finding of a relative deficiency of phospholipids to TG. However, our collective data in S2 cells and SUM cells do not indicate a direct role for seipin in phospholipid synthesis during the earliest stages of LD formation, and we continue to believe that any changes likely reflect later indirect effects of seipin deficiency. We believe that our article now is the first to clearly show that the formation and growth of initial LDs in the ER is a protein-facilitated process, and to identify seipin as an essential player in mediating a step in this process, namely the conversion of nascent iLDs to mature iLDs.